# Engineering polar nanoclusters for enhanced microwave tunability in ferroelectric thin films

Hanchi Ruan[1,4], Hangfeng Zhang [1,4], Vladimir Roddatis[2], Subhajit Pal[3], Joe Briscoe [3], Theo Graves Saunders [1], Xuyao Tang [3], Haixue Yan [3] ✉ & Yang Hao [1] ✉

Microwave tunable thin films that can dynamically adjust dielectric properties are essential for next-generation communication and sensing technologies. However, achieving high-tunability often comes at the cost of increased dielectric loss or the need for large bias electric fields. In this study, we address this challenge by engineering nanoclusters in a tin doped barium titanate thin film and systematically investigate their polarization behaviour across the ferroelectric–paraelectric transition. The optimized film exhibits outstanding microwave tunability (~74% at 6 GHz under a low DC bias of 15 V), which are attributed to the presence of polar nanoclusters embedded within a macroscopically non-polar cubic matrix, stabilized by subtle structural features such as twin boundaries, local lattice distortions, and compositional variations. Structural and dielectric analyses confirm that these nanoclusters remain active, enabling strong field-induced permittivity modulation near room temperature. This work demonstrates a promising strategy to achieve high tunability with minimal losses in ferroelectric thin films, thereby addressing a key performance trade-off in the design of advanced microwave tunable devices.

Tunable microwave dielectric materials are increasingly being explored to meet the stringent requirements of modern telecommunication systems[1]. Among these, ferroelectric materials, known for their electric field-dependent dielectric permittivity, are particularly attractive for electrically tunable applications. As temperature increases, a ferroelectric material undergoes a transition from a polar ferroelectric (FE) phase to a macroscopically non-polar paraelectric (PE) phase above its Curie point ($T_c$). Notably, even in the centrosymmetric PE phase, microscopic short-range polarization can persist in the form of polar nanoclusters embedded in a nonpolar cubic matrix[2]. These polar nanoclusters can be reoriented by an external electric field, enabling tunable dielectric properties[3]. Recent studies have demonstrated that BaTiO$_3$ nanocrystals can simultaneously exhibit both the centrosymmetric cubic and non-centrosymmetric tetragonal phases[4,5]. Additionally, growing evidence suggests that ferroelectric materials in their paraelectric phase contain polar nanoclusters[6,7]. While these nanoclusters appear to enhance dielectric tunability, the details of polarization mechanisms in the PE phase and their contribution to tunability remain unclear. Moreover, developing high-performance tunable dielectrics is often challenged by the inherent trade-off between achieving high tunability, minimizing dielectric loss, and maintaining low bias electric fields, particularly near room temperature.

High tunability is desirable for efficient frequency modulation, but it is often accompanied by increased dielectric loss, which degrades the overall performance of microwave devices[8]. Additionally,

[1]School of Electronic Engineering and Computer Science, Queen Mary University of London, Mile End Road, London, UK. [2]GFZ Helmholtz Centre for Geosciences, Potsdam, Germany. [3]School of Engineering and Materials Science, Queen Mary University of London, London, UK. [4]These authors contributed equally: Hanchi Ruan, Hangfeng Zhang. ✉e-mail: h.x.yan@qmul.ac.uk; y.hao@qmul.ac.uk

achieving a large dielectric tunability typically requires a high applied electric field, particularly in bulk ferroelectrics, which can lead to increased power consumption, reliability concerns in practical applications[9]. These interdependent factors underscore the complexity of optimizing tunable dielectric materials for practical microwave applications, where stable performance must be maintained under ambient operating conditions.

Barium titanate ($BaTiO_3$) based ferroelectrics have attracted significant interest for tunable microwave applications due to their exceptional dielectric response under electric fields. Substituting $Ti^{4+}$(0.605 Å) with larger ions such as $Sn^{4+}$(0.690 Å), has proven to be an effective strategy to modify the ferroelectric-to-paraelectric phase transition, tailoring both permittivity and loss properties[10–13]. Compared with $Ba_{1-x}Sr_xTiO_3$, $BaTi_{1-x}Sn_xO_3$ exhibits superior dielectric tunability at radio frequency, primarily due to the greater chemical stability of Sn compared to Ti ion[14,15]. Previous studies have shown the Sn doped barium titanate exhibits normal ferroelectric for $x \leq 0.175$ and transforms to relaxor ferroelectric state for $x \geq 0.30$, respectively[16,17]. In the intermediate crossover region ($0.175 < x < 0.25$), features of both ferroelectric and relaxor behaviours can coexist, attributed to the formation of polar nanoregions[18]. As the Sn content increases, the transition from a normal ferroelectric to relaxor ferroelectric phase leads to a decrease in tunability value of materials[17]. Notably, $BaTi_{0.85}Sn_{0.15}O_3$ exhibits a Curie point near room temperature[19,20], placing it just below the crossover region and making it highly suitable for practical microwave tunable applications. This near-room-temperature $T_c$ enables the formation of polar nanoclusters within an average cubic paraelectric matrix, significantly enhancing the electric field-induced dielectric tunability while remaining low loss.

Compared with bulk ceramics, $BaTi_{1-x}Sn_xO_3$ thin films exhibit moderate dielectric permittivity, striking an attractive balance between enhanced electronic performance and device miniaturization[21–24]. In this study, (BTS) thin films were fabricated via a sol–gel spin-coating process, and their structural, microstructural, and dielectric properties across different sintering temperatures were systematically investigated. Structural analyses confirmed a cubic perovskite structure for all films, while microstructural characterization revealed that the BTS thin film sintered at 850 °C exhibits an optimal balance between tunability and dielectric loss, attributed to the presence of polar nanoclusters near $T_c$. Our findings provide compelling evidence that local polarization structures, such as polar nanoclusters, play a crucial role in enhancing microwave tunability. These nanoclusters enable significant dielectric permittivity modulation under an applied electric field. This work advances the understanding of polarization dynamics in BTS thin films and offers new strategies for optimizing tunable dielectric materials for microwave applications.

## Results

### Phase structures and microstructure of BTS thin films

BTS thin films were prepared using a sol–gel and spin-coating on Pt/Ti/$SiO_2$/Si substrates, followed by sintering at 800 °C, 850 °C, and 900 °C, denoted as BTS800, BTS850, and BTS900, respectively. X-ray diffraction (XRD) patterns of BTS films were well fitted with a cubic perovskite structure in the space group $Pm$-$3m$ (Fig. 1a–c), with refinement details summarized in Table S1. Notably, diffraction peak intensities increase with sintering temperature, reflecting improved crystallinity resulting from enhanced atomic mobility at elevated temperature. Furthermore, the lattice parameters of BTS film expand with increasing sintering temperature, which might be a result of enhanced grain growth and reduced internal stress that normally constrains lattice expansion. Scanning electron microscopy (SEM) and Energy Dispersive X-ray (EDX) images of cross-section (Fig. 1d–f and Fig. S2) and surface (Fig. S1) morphology reveal the impact of sintering

temperature on the microstructure and interface characteristics. All BTS films exhibit relatively uniform thicknesses (525 nm for BTS800, 550 nm for BTS850, and 500 nm BTS900), mainly due to a 750 °C high-temperature preheat treatment after each deposition layer, which promotes structural densification and integrity. A few isolated pores are observed, but given their low concentration and large permittivity contrast between air ($\varepsilon' \approx 1$) and BTS ($\varepsilon' \approx 600$), the effect on the overall dielectric constant is expected to be very minor. The surface morphology images show the average grain size increases from $23.6 \pm 5.2$ nm (BTS800) to $69.5 \pm 16.9$ nm (BTS900), with a rise in root-mean-square surface roughness (Rq) from 3.77 nm to 5.08 nm (Fig. S1). The BTS800 film exhibits the smallest grain structure, with some surrounding amorphous regions, minimal interfacial diffusion, and a smooth interface. Increasing the sintering temperature to 850 °C promotes grain growth and reduces grain boundary density. However, further increasing the temperature to 900 °C (BTS900) leads to a significant interfacial diffusion between the BTS film and platinum electrode (Fig. S2), potentially compromising film adhesion and electric performance. The X-ray photoelectron spectroscopy (XPS, Fig. 1g) results reveal pronounced shoulder features in the Ba $3d$ and Sn $3d$ spectra of BTS800, suggesting residual components from incomplete precursor decomposition. These shoulders are significantly weakened in BTS850 and BTS900, consistent with SEM observations of amorphous regions in BTS800 and the well-defined grains at higher sintering temperatures. The O $1s$ spectra show a more intense defect-related peak at $ca.$ 531.5 eV in BTS800, attributed to oxygen vacancies or absorbed/residual oxygen components, which decreases with increasing sintering temperature. Notably, no clear binding energy shifts are observed in the main peaks of measured elements, indicating stable oxidation states and minimal influence of oxygen vacancies.

### Thermal and electric field dependent of dielectric properties

Dielectric properties at low frequency were measured using a large circular gold electrode in conjunction with a platinum bottom electrode layer (Fig. 2a). Figure 2b shows the dielectric permittivity increases with sintering temperature, which is attributed to the reduction in grain boundary density, as confirmed by SEM images (Fig. S1a-c), since grain boundaries are non-ferroelectric component and exhibit low dielectric permittivity[25]. All BTS films exhibit higher dielectric permittivity at lower frequencies, followed by gradually decreasing with increasing frequency. This phenomenon arises from the frequency dependent response of various polarization mechanisms, some of which become inactive at higher frequencies due to their slower relaxation times[26]. Among the samples, BTS900 shows the highest permittivity, followed by BTS850 and BTS800, consistent with enhanced grain growth and improved crystallinity observed in both SEM, EDX, and XRD analyses. The larger grains in BTS900 reduce grain-boundary density and relieve internal stresses, enabling more rapid dipole response and resulting in higher dielectric permittivity. Dielectric loss decreases with increased frequency in Fig. 2b, primarily due to the suppression of the extrinsic loss mechanisms at high frequencies. Around 100 kHz, the dielectric loss exhibits a notable drop with increasing frequency, which might be attributed to the dielectric relaxation associated with dipole polarization[27]. Since the $T_c$ temperature of BTS900 was slightly above room temperature (Fig. S4), dipolar contribution remains active and relax around this frequency range. In contrast, BTS800 and BTS850, with their $T_c$ below room temperature, exhibit no such anomaly, because their dipoles are less active. The drop in loss at high frequencies likely results from the diminished contribution of dipolar polarization. Notably, BTS850 exhibits the lowest dielectric loss at 1 MHz, which may be attributed to its optimal microstructure and reduced interfacial defects. The dielectric tunability at radio frequency improves with increasing sintering temperature, with BTS900 demonstrating the highest tunability, in agreement with its high permittivity (Fig. 2c). All samples

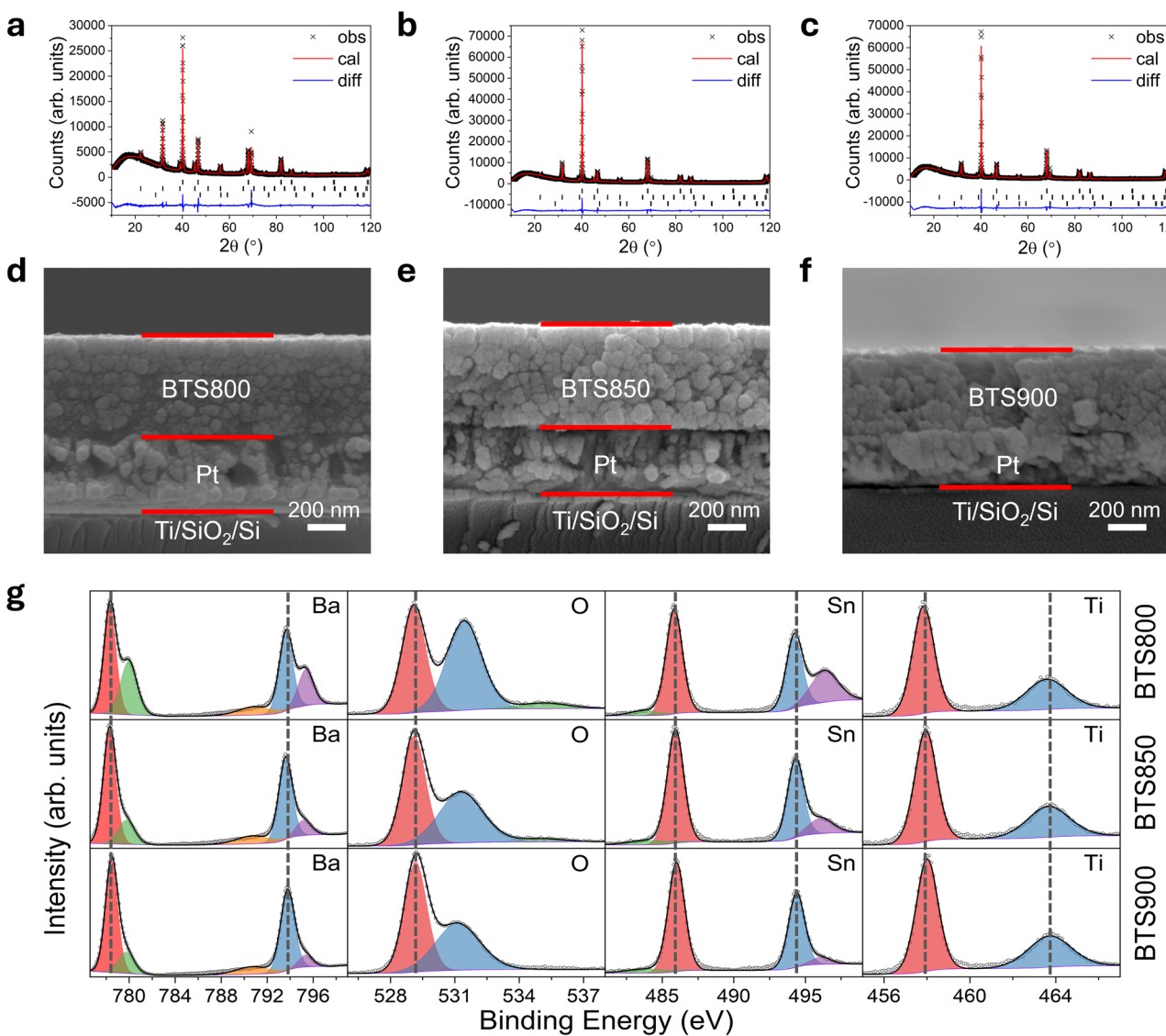

**Fig. 1 | XRD characterization of the BTS thin films.** Fitted XRD profiles for **a** BTS800 thin film, **b** BTS850 thin film, and **c** BTS900 thin film, with Bragg peak positions indicated by black markers for Pt (upper), BTS (middle), and Si (lower). The cross-sectional SEM images of **d** BTS800 thin film, **e** BTS800 thin film, and **f** BTS900 thin film grown on the Pt/Ti/SiO$_2$/Si substrate. **g** Fitted XPS spectra of Ba 3 d, O 1 s, Sn 3 d, and Ti 2p for studied films.

exhibit non-linear polarization-electric field (*P-E*) hysteresis loops, accompanied by two current peaks in the corresponding current-electric field (*I-E*) loops (Fig. S3). While the broadened *P-E* loops and tilted *I-E* loops suggest the presence of leakage current, possibly induced by the space charges, the displacement of switching current peaks from the maximum applied field indicates polar nanoclusters are responsible for the polarization behaviour in the film[28]. The sharpest current peaks are observed in BTS900, further supporting its superior tunability. Structurally, pure barium titanate typically exhibits a tetragonal phase with an elongated *c*-axis, associated with the off-centre displacement of Ti$^{4+}$ ions within the oxygen octahedra, resulting in long-range ferroelectric ordering. In the present system, partial substitution with Sn$^{4+}$ disrupts this ordering, leading to the formation of polar nanoclusters embedded within a non-polar cubic matrix. This structural modification significantly influences overall dielectric and ferroelectric behaviour of the BTS films.

Unlike BTS bulk ceramics, which exhibit sharp dielectric permittivity peaks at the Curie point[29], the thin films show a broader permittivity peak, which gradual increase in dielectric permittivity upon heating (Fig. S4), followed by a subsequent decrease. A subtle anomaly

near −20 °C is observed in the BTS800 sample, likely arising from the nanoscale coexistence of the multiple polar phases within nanograins[30,31]. The observed $T_c$ temperatures are 3.8 °C, 16.4 °C, 23.2 °C for BTS800, BTS850 and BTS900, respectively. The shift of $T_c$ to higher temperatures is attributed to the combined effects of the grain size, crystallinity and the internal stress within the films[32,33]. During the paraelectric-ferroelectric phase transition at $T_c$, *c*-axis elongation leads to local lattice distortions and development of internal stress within grains. In multidomain grains, such internal stress can be partially relieved through non-180° domain walls (twin boundaries), whereas in single-domain grains, stress relaxation is constrained. This restriction suppressing the *c/a* ratio and reduces $T_c$[34,35]. Increasing sintering temperatures promotes grain growth, which reduces internal stress and thereby increases $T_c$, consistent with observations reported by Song et al.[32] and Chen et al.[36]. Moreover, the frequency independence of $T_c$ of BTS thin films indicates the absence of relaxor type of ferroelectric behaviour. It's worth noting that barium titanate based relaxor ferroelectrics (such as BaTi$_{0.90}$Ga$_{0.05}$O$_3$ and BaTi$_{0.70}$Zr$_{0.30}$O$_3$)[37] also exhibit high tunability, but they suffer higher microwave loss than the normal ferroelectrics such as Ba$_{0.6}$Sr$_{0.4}$TiO$_3$.

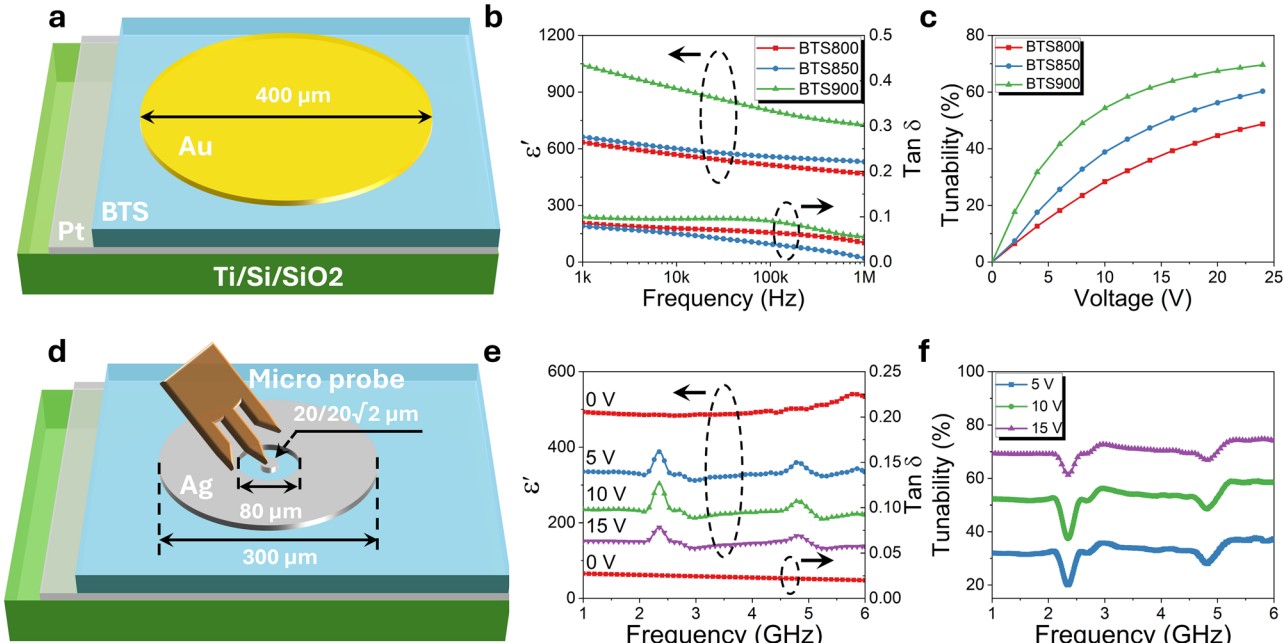

**Fig. 2 | Dielectric properties of the BTS thin films.** Schematic of the dielectric tunability measurement setup covering both **a** low and **d** high frequency ranges. **b** frequency dependence of dielectric permittivity and loss tangent for studied BTS films in the low frequency range under a 0.5 V AC voltage. Electric field dependent of **c** dielectric tunability for BTS films measured at 1 MHz. **e** Dielectric permittivity and **f** tunability for the BTS850 film in the high frequency range. convert voltage to field. (note: A DC bias of 1 V corresponding to an electric field of ca. 2 kV·cm⁻¹).

Compared to the BTS800, the BTS850 thin film exhibits a $T_c$ closer to room temperature, resulting in a higher concentration of polar nanoclusters and enhanced dielectric tunability. In contrast, the BTS900 thin film shows a broader phase transition above room temperature, suggesting the effect of the rough interface, which also contribute to increased dielectric loss. These observations highlight the inherent trade-off in BTS system, where higher permittivity and tunability can be achieved at cost of higher dielectric loss. BTS850 achieving an optimal balance high tunability and low dielectric loss, due to its proximity to room temperature $T_c$ and reduced interface-related dielectric losses.

Since most practical applications operate in the GHz range, it is essential to evaluate the dielectric and tunable properties at microwave frequencies[38]. For high-frequency measurement, a specialized electrode configuration was employed, consisting of a large silver ring electrode encircling a small central circular silver patch (Fig. 2d). Two electrode patterns with different central patch radii (20 and 20√2 μm), but identical outer ring dimensions, were used to minimize parasitic effects and ensure accurate extraction of microwave dielectric properties. At 1 GHz, the extracted dielectric permittivity was *ca.* 493, closely matching the dielectric permittivity of 531 at 1 MHz (531), indicating that polar nanoclusters remain active and responsive at microwave frequencies. The dielectric spectrum of BTS850 shows relatively stable permittivity at zero electric field across the frequency range from 1 GHz to 6 GHz, further supports the contribution of polar nanocluster at microwave frequency (Fig. 2e). The dielectric loss of BTS850 was found to be *ca.* 0.027–0.020 across the 1–6 GHz frequency range, which is comparable to the other tunable thin films[27,39–43] (Table S4). The dielectric permittivity decreased uniformly across the frequency range under electric field. Additionally, resonant peaks were only observed at 2.3 GHz and 4.8 GHz under bias fields, attributed to piezoelectric effect[44–47], where aligned dipoles induce resonance phenomena at specific frequencies. Excluding these resonance regions, the BTS850 thin film exhibits a high tunability of *ca.* 74% under a relatively low applied DC bias of 15 V (equivalent of 273 kV cm⁻¹) with a low loss of 0.020, yielding a figure of merit (FoM) of 37 at 6 GHz (Table S4).

## Investigation of polar nanoclusters and their polarization mechanisms

Piezoresponse force microscopy (PFM) was used to reveal distinct polarization behaviour of the polar clusters in the BTS850 thin film across its ferroelectric and paraelectric phases. At 14 °C, which is below its Curie point $T_c$, the film exhibits a uniform phase contrast in the PFM image (Fig. 3a), with ferroelectric domains occurring multiple grains, indicative of well-defined and long-range ferroelectric ordering. Upon application of a DC bias of ±10 V, clear domain switching is observed (Fig. 3b), further supporting the existence of switchable polarization. The phase and amplitude hysteresis loops (Fig. 3c) show sharp, square-like switching behavior, characteristic of long-range ferroelectric domains. At 21 °C, the film transitions into the paraelectric phase, where the phase contrast becomes weaker (Fig. 3d). However, a residual polarization response to applied DC bias remains observable (Fig. 3e), suggesting the presence of polar nanoclusters rather than coherent long-range ferroelectric domains. These randomly oriented polar nanoclusters contribute significantly to dielectric tunability, as they can readily rotate and aligning under external electric fields. The persistent phase contrast after removal of the DC bias indicates the existence of relaxation processes, which are responsible for the hysteresis observed in the *P-E* loops at 20 °C (Fig. S3). The grain size distribution obtained from SEM analysis (Fig. S1h) closely matches the domain size distribution observed in the PFM amplitude images (Fig. S5-6), confirming a single-grain, single-domain configuration in the ferroelectric phase of the BTS850 thin film.

Raman spectra were deconvoluted into seven distinct vibrational modes using Lorentzian peak fitting (Figs. 3f and S7). These modes can be categorised based on wavenumber ranges: low-frequency (<200 cm⁻¹), mid-frequency (200–400 cm⁻¹), and high-frequency (>400 cm⁻¹) regions. The low-frequency modes are primarily linked to lattice vibrations, while mid-frequency peaks correspond to vibrations of B–O bonds. The high-frequency modes (>400 cm⁻¹) are associated with stretching and breathing vibrations in the $BO_6$ octahedra. Notably, weak Raman peaks at *ca.* 300 cm⁻¹ and 730 cm⁻¹ (Fig. 3f) correspond to local ferroelectric tetragonal polar structure

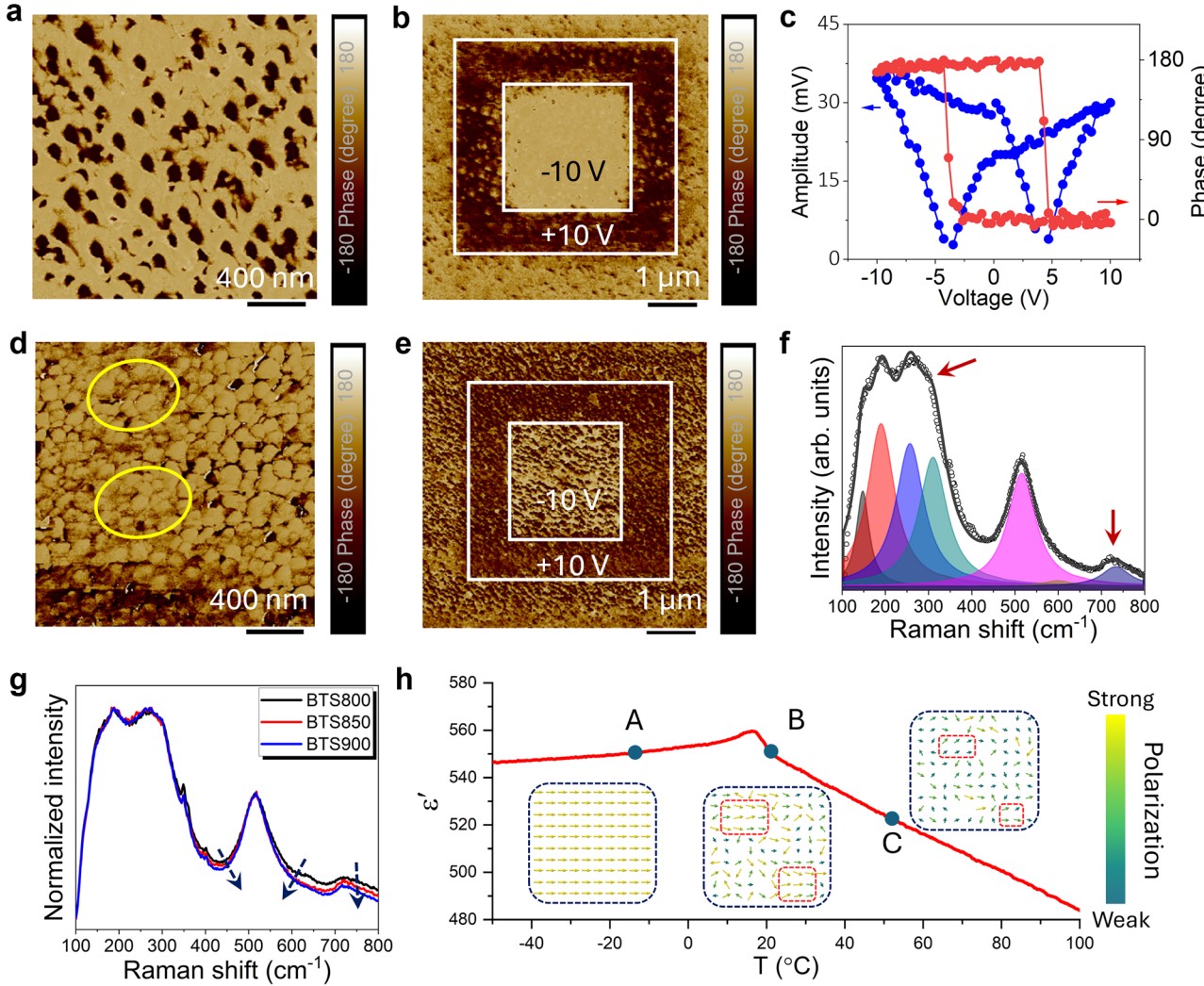

**Fig. 3 | PFM and Raman analysis of the BTS850 thin film. a–c** were measured below its Curie temperature at 14 °C, while **d–f** were measured above its Curie temperature at 21 °C. **a** PFM phase image of the BTS850 thin film measured at its ferroelectric phase. **b** Phase-contrast PFM images demonstrating domain switching inside the BTS850 thin film, which arise after applying a ± 10 V DC bias to the tip. **c** Phase and amplitude switching spectroscopy loops of the BTS850 thin film, demonstrating ferroelectric hysteresis. **d** PFM phase image of the BTS850 thin film measured at its paraelectric phase. The signature of domain merging can be observed from the circles. **e** Phase-contrast PFM images of the BTS850 thin film with opposite polarization direction after applying a ± 10 V DC bias to the tip. **f** Fitted Raman spectrum of the BTS850 thin film and **g** Raman spectra of all BTS samples with the arrow represent the variation of the peak profile. **h** Temperature-dependent dielectric permittivity of the BTS850 thin film measured at 100 kHz from 0 °C to 50 °C. The inset figures present the temperature dependency of the alignment structure of polarizations, where the arrows show the polarization profile.

within the nominally cubic matrix, indicating the presence of polar nanoclusters even after the disappearance of macroscopic ferroelectricity[48,49]. Additionally, BTS800 shows slightly broader Raman peaks at the *ca.* 520 cm⁻¹ and 730 cm⁻¹ compared to BTS850 and BTS900 (Fig. 3g), which could arise from several factors such as local lattice distortion, local ferroelectric ordering involving $Sn^{4+}$ and $Ti^{4+}$ ions, and residual impurity in the system[50,51]. Higher sintering temperature enhances the local ferroelectric ordering, leading to sharper Raman features and improved ferroelectric characteristics.

Based on these findings, the evolution of polarization states with temperature in BTS850 is schematically summarized in Fig. 3h. The inset images in Fig. 3h illustrate a qualitative visualization of this progression from ferroelectric domains to polar nanoclusters on heating. Below $T_c$ (point A), dipoles exhibit long-range ordered states, forming well-defined domain structure with uniform polarization within individual grains. These domains can be switched by applying external DC fields. At temperature slightly above $T_c$ (point B), the long-range order collapses into a state populated by randomly oriented polar nanoclusters, though localized regions may still exhibit partial dipole alignment[10]. As temperature rises further, the size of these polar nanoclusters decreases, leading to a reduced dielectric permittivity and tunability. Consequently, the BTS800 thin film, measured further from its $T_c$, shows lower dielectric permittivity and tunability compared to BTS850, which is characterized near its optimal temperature range, as shown in Fig. 2c.

Figure 4 shows a comprehensive analysis of the microstructure and crystallographic characteristics of BTS850 using high-resolution transmission electron microscopy (HRTEM) and fast Fourier transform (FFT). The high-angle annular dark field (HAADF) image taken along [001] crystallographic direction reveals an ordered atomic arrangement with two distinct regions marked by orange squares (Fig. 4a). FFT analysis of these regions (Figs. 4b and c) shows additional diffraction spots (indicated by arrows), which are slightly displaced from the ideal diagonal positions denoted by dotted lines. This displacement is indicative of an incommensurate phase induced by local lattice distortions and is associated with the formation of polar nanoclusters. Local dislocations were observed in high resolution HAADF and ADF images (Fig. S8a, b), indicating nanoscale heterogeneity within the

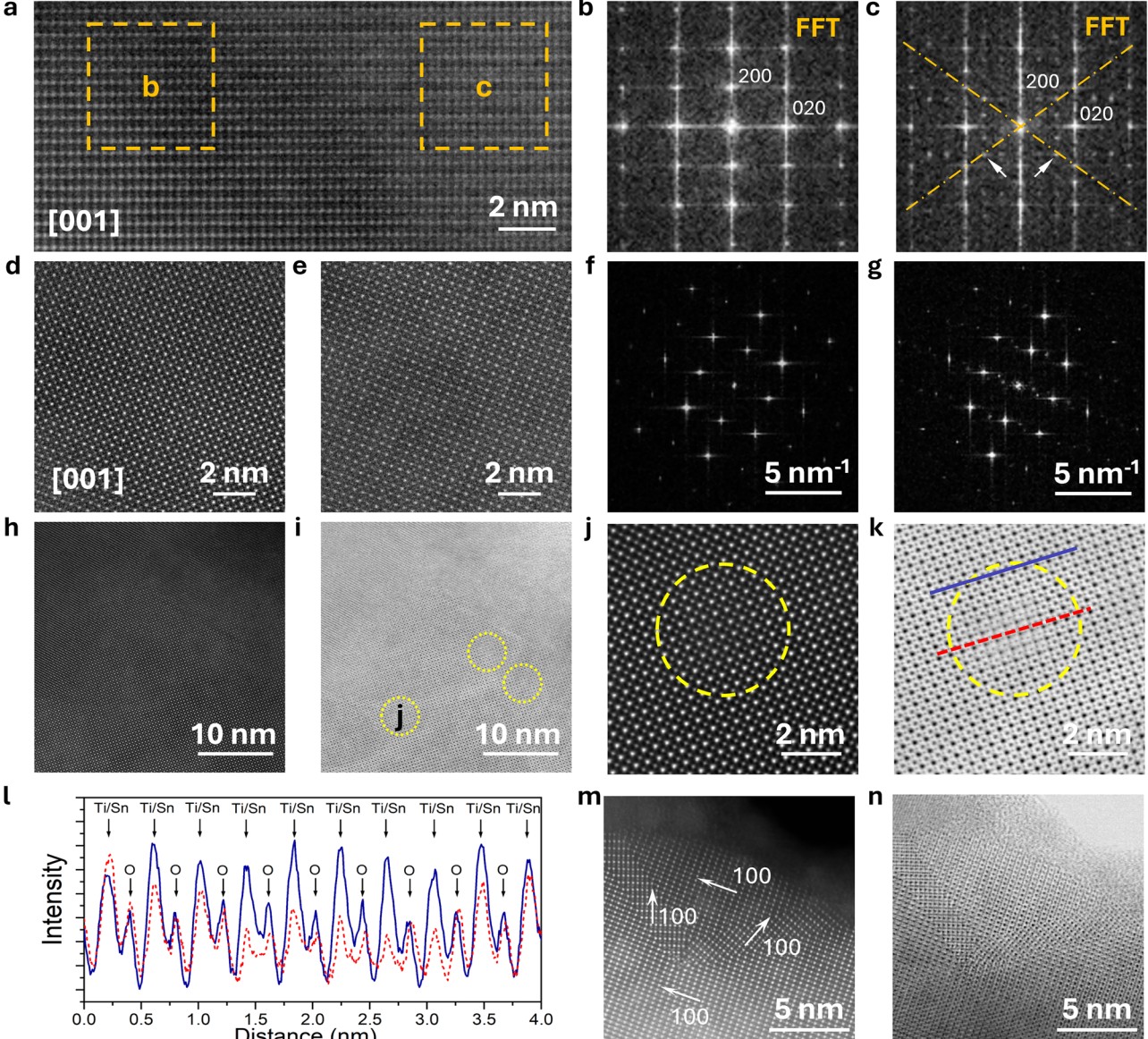

**Fig. 4 | STEM characterization results. a** high-resolution HAADF image of BTS850 along [001] crystallographic direction with corresponding **b** and **c** FFTs taken from selected area with orange dashed square. Two extra spots are marked with white arrowheads. **d, e** High-resolution HAADF images from different areas of the sample, corresponding to the FFT patterns shown in (**f, g**), respectively. **h** HAADF and **i** ADF of grain along [001] BTS850 crystallographic direction; a few nanodomains are marked with dashed circles. **j** HAADF and **k** ADF of the enlarged images of selected domain (yellow dashed circle), and corresponding line profiles (red dashed line and indigo solid line in k) shown in **l** reveal the reduced intensity of Ba and Ti/Sn atomic columns compared with similar columns in the surrounding areas. **m** HAADF and **n** ADF of a grain imaged along [110] with multiple twinning, showing parts with three different [100] orientations.

BTS850 matrix and suggesting the presence of polar nanoclusters. Additionally, HAADF images across different regions and their corresponding selected area electron diffraction (SAED) patterns (Fig. 4d–g, Fig S8c and d) exhibit some additional diffraction spots, further supporting the existence of local structural distortions and short-range ordering in the system. Elemental maps of O, Ti, Sn and Ba (Fig. S9-10) across the interface between BTS850 and Pt electrode demonstrate a uniformly dispersed distribution of all constituent elements across the imaged region. The roughness of BTS850/Pt interface can be estimated to be about 15 nm (Fig. S9). The corresponding chemical mapping did not reveal significant diffusion of Pt into the BTS850 film. The semi-quantitative energy-dispersive X-ray (EDX) data are provided in Tables S2 and S3. Detailed examination of HAADF and ADF images reveals the presence of nanoclusters (marked by yellow circles) ranging from 2 to 10 nm in diameter (Fig. 4h, i). High-magnification images of these nanoclusters (Fig. 4j, k) offer detailed insight into structure,

with ADF imaging allowing visualization of all atomic columns, including oxygen. Intensity profiles extracted along the black and red dashed lines shown in Fig. 4k, reveal noticeable intensity reduction in Ba and Ti/Sn atomic columns, as well as positional shifts in oxygen atomic columns (Fig. 4l). Thickness analysis performed using electron energy loss spectroscopy (EELS) confirms identical values within and outside the nanoclusters, thus excluding the possibility of voids induced intensity variations. Therefore, this intensity reduction suggests that within these domains the atoms in all atomic columns are slightly displaced from their ideal positions as well as subtle compositional variation, resulting in less coherent electron scattering and reduced atomic column intensities. Furthermore, Fig. 4m, n shows the presence of multiple twin boundaries within a single grain in the BTS850 film. The coexistence of these twin boundaries, local lattice distortions, polar nanoclusters, and subtle compositional variations

contributes to the stabilization of polar nanoclusters and influences the observed polarization behaviour.

## Discussion

In this work, a comprehensive analysis of polarization behaviour in BTS thin film fabricated via sol-gel spin coating route across both the ferroelectric and paraelectric phases was conducted. An exceptional microwave dielectric tunability of *ca.* 74% at 6 GHz under a low 15 V DC bias field. The remarkable performance is attributed to the engineering of polar nanoclusters within the thin film matrix. These nanoclusters are embedded in a macroscopically non-polar cubic structure and are stabilized by subtle structural features including local lattice distortions, twin boundaries, and nanoscale compositional inhomogeneities. These polar nanoclusters exhibit a robust field responsive character, enabling strong field induced permittivity modulation to achieve maximal tunable response. These findings establish a novel design strategy for high-performance tunable materials, illustrating how engineered polar nanoclusters and tailored phase transition temperature can be leveraged to realize device-relevant microwave performance in ferroelectric thin films.

## Methods

### Preparation of precursor solution

In this process, barium acetate [Ba $(CH_3COO)_2$], titanium *n*-butoxide $(Ti[O(CH_2)_3CH_3]_4)$ and tin 2-ethylhexanoate $(C_{16}H_{30}O_4Sn)$ were used as starting materials. The glacial acetic acid $(CH_3COOH)$ and 2-methoxyethanol $(CH_3OCH_2CH_2OH)$ were used as the solvent. The acetylacetone $(AcAc, CH_3COCH_2COCH_3)$ was used as the polymerizing agent. Firstly, barium acetate was dissolved in the acetic acid under stirring for 2 h (solution 1) at room temperature. Secondly, 2-methoxyethanol and AcAc were mixed following a predetermined ratio, and then a stoichiometric amount of titanium *n*-butoxide was added under continuous stirring for 0.5 h at room temperature. The molar ratios of Ti:AcAc and Ti:2-methoxyethanol were 1:2 and 1:5, respectively. Then tin 2-ethylhexanoate was added to the solution under stirring (solution 2). Thirdly, solution 1 was mixed with solution 2, and then acetic acid was added to control the concentration of the solution to 0.35 M. Finally, the mixed solution was stirred for 4 h. After ageing for 24 h, the prepared $BaTi_{0.85}Sn_{0.15}O_3$ precursor solution can be used for depositing film layers.

### Film fabrication

The prepared $BaTi_{0.85}Sn_{0.15}O_3$ precursor solution was deposited on the Pt/Ti/SiO$_2$/Si substrates by spin coating at a speed rate of 3000 rpm for 30 s for each layer. The sol-gel deposited thin films were preheated at 200 °C, 450 °C, and 750 °C for 5 min, respectively, to evaporate the solvent and decompose the organic residuals. The preheat with a relatively high temperature of 750 °C could help remove undesired organic residuals more completely, thus the BTS thin films could exhibit denser microstructure. The above processes were repeated six times to obtain the preheated thin films with a certain film thickness. Finally, these preheated BTS thin films were annealed at certain high temperatures of 800 °C, 850 °C, or 900 °C with a heating rate of 5 °C/min for 15 min to obtain the final crystallized thin films. The shorter annealing time induced a lower diffusion process of Pt and Ti in the BTS thin film.

### XRD, SEM, and XPS characterization

To investigate the crystal structure of the BTS thin films, X-ray diffraction (XRD) analysis was performed using a PANalytical X'Pert Pro diffractometer with an X'Celerator detector, employing Ni-filtered Cu-Kα radiation ($\lambda = 1.5418$ Å). For morphological analysis of both surface and cross-section, the films were sputter-coated with gold or carbon and imaged using a field-emission scanning electron microscope (FE-SEM, FEI Inspect-F Oxford) at an accelerating voltage of 10 kV. X-ray

photoelectron spectroscopy (XPS, ThermoFisher Nexsa X-ray Photoelectron spectrometer) was used to characterize the surface speciation.

### PFM imaging

The PFM measurements were carried out using a Bruker ScanAsyst Dimension AFM in the contact resonance mode using platinum and iridium-coated tip (SCM-PIT-V2, Bruker) with a force constant of *ca.* 3 N/m. The PFM imaging was performed by applying a 3 V AC to the tip with a 346 kHz driving frequency and −82° drive phase offset. The amplitude fitting of the quality factor (Q) and coefficient of determination ($R^2$) is 98.2 and 0.98. To obtain a PFM switching image, +10 and −10 V DC were applied through the AFM tip to pole the ferroelectric surface on the 4 × 4 and 2 × 2 μm$^2$ area, respectively. Finally, after removing the DC voltage, the PFM image was captured in 6 × 6 μm$^2$ with 2 V AC drive amplitude. The phase and amplitude loops were extracted in the SS-PFM mode. The PFM phase and amplitude loops were extracted in the "off-field" segments using Python script.

### TEM characterisation

Transmission electron microscopy (TEM), including electron diffraction (ED), high-angle annular dark field (HAADF), and scanning TEM (STEM) studies, was performed using a Thermo Fisher Scientific Themis Z 80–300 Cs-aberration probe-corrected electron microscope operated at 300 kV and equipped with Super-X Energy Dispersive X-ray EDX detector, and a Gatan Continuum 1065ER Electron Energy Loss Spectrometer. TEM specimens were prepared using a Thermo Fisher Scientific Helios G4UC focused ion beam instrument operated at 30 kV, 16 kV, and 5 kV. The final cleaning was performed using a Gatan DuoMill 600 device.

### Dielectric measurement below 1 MHz

Dielectric measurements of the fabricated BTS thin films were characterized in the low-frequency range (up to 1 MHz), using a circular gold top electrode (0.4 mm in diameter), which was deposited through a custom-designed mask via the EMITECH SC7620 sputter coater. The resulting top gold electrode, the BTS film and the underlying platinum electrode formed a standard metal–dielectric–metal (MDM) capacitor. Electrical characterization, including dielectric permittivity, tunability, and loss tangent, was performed across a frequency range of 1 kHz to 1 MHz using an Agilent 4294 A precision impedance analyzer. To examine the temperature dependence of dielectric behavior, measurements were carried out from −30 °C to 50 °C using an Agilent 4284 A LCR meter. Ferroelectric properties, including current–electric field (*I–E*) and polarization–electric field (*P–E*) loops, were characterized using a ferroelectric hysteresis tester (NPL, UK). A triangular voltage waveform at 10 Hz was applied to evaluate the ferroelectric switching response.

### Microwave measurement

To fabricate patterned parallel-plate electrodes for microwave testing, a silver layer was sputter-deposited onto the film surface using a magnetic mask alignment method. The mask, made from 430 stainless steel (a magnetic alloy), was securely held against the BTS850 thin film sample by magnetic force from a base holder underneath the substrate. This configuration with mask, film, and holder formed a stable sandwich structure during sputtering using an Agar Auto coater. The even magnetic attraction ensured precise pattern transfer onto the film surface.

After electrode patterning, one-port reflection measurements in the microwave frequency range were conducted using an Agilent N5230C PNA-L Network Analyzer under various DC bias conditions. The corresponding scattering parameters (*S* parameters) were recorded for further analysis. A 100 μm ground–signal–ground (GSG) RF probe (model 40M-GSG-100-PLL, GGB Industries) connected with the

PNA-L was used to contact the patterned electrode. To simultaneously apply DC bias and measure the high-frequency signal, a bias-tee (model SB18D3D, SigaTek) was used during the testing.

## Extraction of microwave dielectric properties

The measured reflection coefficients $S_{11}$ could be converted to the impedance of the device under test (DUT) by using a simple relationship[52,53]:

$$Z_{DUT} = Z_0 \frac{1 + S_{11}}{1 - S_{11}} = R + jX \tag{1}$$

where $Z_{DUT}$ is the impedance of the device under test (DUT), and $Z_0$ is the characteristic impedance (50$\Omega$) of the transmission cable. The test structure here is not a simple and ideal parallel plate capacitor. The dielectric permittivity and loss tangent of the measured ferroelectric thin films cannot be simply calculated from the measured capacitance value and loss of the test structure by using the thickness of the thin film and the area of the center silver patch [38]. The capacitance between the bottom platinum layer and the top ground affects the accuracy of the measurement. The resistance in the top and bottom electrodes causes additional loss in the test structure. Taking into account these conditions and assuming the feature sizes of the test structure are much smaller than the wavelength of the microwave signal, the test structure could be analyzed following an equivalent circuit as shown in Fig. S7 [39]. $R_{center}$, $R_{outer}$, $C_{center}$, $C_{outer}$ represent the shunt resistance and the capacitance of the ferroelectric thin film layer between the bottom electrode and the top electrodes, referring to the centre circular patch and the external electrode. $R_{ring}$ corresponds to the resistance in the bottom electrode between the circular patch and the external electrode. Thus, the $Z_{DUT}$ from the circuit as shown in Fig. S8 can be expressed as

$$Z_{DUT} = R_{center} + R_{outer} + R_{ring} + \frac{1}{j\omega}\left(\frac{1}{C_{center}} + \frac{1}{C_{outer}}\right) \tag{2}$$

Equation (2) enables to obtain the dielectric permittivity of the ferroelectric thin film by measuring two different parallel plate varactors with different inner electrode radii ($r_1$, $r_2$) but the same outer electrode radius ($R$). By subtracting the imaginary part of the $Z_{DUT}$ of these two structures, the parasitic effect could be removed:

$$Im(Z_{DUT}^1) - Im\left(Z_{DUT}^2\right) = \frac{1}{j\omega}\left(\frac{1}{C_{center}^1} + \frac{1}{C_{outer}^1}\right) - \frac{1}{j\omega}\left(\frac{1}{C_{center}^2} + \frac{1}{C_{outer}^2}\right) \tag{3}$$

where $Im(Z_{DUT}^1)$ and $Im(Z_{DUT}^2)$ are the imaginary part of the measured impedances of the two test structures. $C_{outer}^1$ and $C_{outer}^2$ are the same since the areas of external electrodes in these two test structures are same, so Eq. (3) could be further simplified as:

$$Im(Z_{DUT}^1) - Im\left(Z_{DUT}^2\right) = -\frac{1}{\omega}\frac{d}{\pi\varepsilon_0\varepsilon'}\left(\frac{1}{r_1^2} - \frac{1}{r_2^2}\right) \tag{4}$$

where $d$ is the film thickness, $\omega$ is the angular frequency ($\omega = 2\pi f$), $\varepsilon_0$ is the vacuum dielectric constant with the value of $8.854 \times 10^{-12}$ F/m and $\varepsilon'$ is the dielectric permittivity of the measured ferroelectric thin film. Thus, $\varepsilon'$ could be calculated as:

$$\varepsilon' = -\frac{d\left(\frac{1}{r_1^2} - \frac{1}{r_2^2}\right)}{\omega\pi\varepsilon_0\left[Im(Z_{DUT}^1) - Im(Z_{DUT}^2)\right]} \tag{5}$$

After extracting the dielectric permittivity of the ferroelectric thin films from the measured $S_{11}$ at a certain applied DC bias voltage, the tunability of the ferroelectric thin films at microwave frequency can be calculated. For the ring section, it only includes the bottom platinum layer, thus, it could be treated as a pure resistance. Since the resistance per square of the platinum layer is low and the areas of this ring section of the two test structures are very close, the $R_{ring}$ value could be assumed as the same value in both two test structures with different central patches. Similarly, the loss tangent of the thin film could be derived by comparing the two test structures:

$$\tan\delta = -\frac{R_1 - R_2}{X_1 - X_2} \tag{6}$$

## Data availability

All data needed to evaluate the conclusions in the paper are present in the paper and/or the Supplementary Materials. Any other relevant data is also available from the corresponding authors upon request.

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

## Acknowledgements

This work is supported by the "Software Defined Materials for Dynamic Control of Electromagnetic Waves" (ANIMATE) project (Grant No. EP/R035393/1), and the authors acknowledge Engineering and Physical Sciences Research Council (EPSRC). The authors thank the European Regional Development Fund and the State of Brandenburg for the Themis Z microscope (part of the Potsdam Imaging and Spectral Analysis (PISA) facility). SP and JB acknowledge European Research Council (ERC) under the European Union's Horizon 2020 research and innovation programmes (Grant No. 101001626).

## Author contributions

H.R. and H.Z. prepared the materials, conducted the materials characterisation and data analysis. V.R., S.P., T.S., and X.T. assisted with materials characterisation and data analysis. H.R., H.Z., J.B., H.Y., and Y.H. contributed to the discussion of results. H.R. and H.Z. wrote the original draft, and all authors contributed to editing the final version of the manuscript. Y. H. acquired the funding and supervised the project.

## Competing interests

The authors declare no competing interests.
