## [Transparent Peer Review file · Nature Communications]

Engineering polar nanoclusters for enhanced microwave tunability in ferroelectric thin films

Corresponding Author: Dr Haixue Yan

Version 0:

Reviewer comments:

Reviewer #1

(Remarks to the Author)

The experimental part of this paper is well done and well documented. However, the scientific part contains a conceptual shortage:

1) The aim of this work is described by the authors as following: "This work advances the understanding of polarization dynamics in BTS thin films and offers new strategies for optimizing tunable dielectric materials for microwave applications", "The remarkable performance (tunability) is attributed to the engineering of polar nanoclusters within the thin film matrix" and "This work demonstrates a promising strategy to achieve high tunability with minimal losses in ferroelectric thin films". On the other hand it is well known that "the tunability in relaxor ferroelectric exhibits a higher value as compared to normal ferroelectrics. It happens due to the presence of short-range ordering PNRs, which respond very quickly with the applied electric field compared to the long-range ordering of dipoles in normal ferroelectrics. [DOI: 10.5772/intechopen.96185]" Thus, the concept of this work is not new.

Also, this work is not self-consistent concerning the presence of polar nanoregions (PNRs) and the based on these PNRs relaxor behavior:

1) Lets start with the definition of relaxors: "Relaxors are defined as ferroelectrics in which the maximum in the temperature dependence of static susceptibility occurs within the temperature range of dielectric relaxation, but does not coincide with the temperature of singularity of relaxation time or soft mode frequency" [DOI: 10.1142/S2010135X1241010X]. This means that a diffuse peak of dielectric permittivity and a typical frequency dispersion in the temperature range of the peak are characteristic features of ferroelectric relaxors.

In this work, figures S3 do not show the sharp peak in the temperature dependence of dielectric permittivity known for relaxors. Also no dispersion of losses above temperature of maximum dielectric permittivity was obtained. Thus, nor dielectric permittivity nor losses show typical behaviour of relaxor ferroelectrics. Finally, the authors themselves exclude relaxor-type of ferroelectric behaviour (row 143-144). This also coincides with the fact that in figures S2a-c hysteresis curves are not the slim ones known for relaxor ferroelectrics. Once the fabricated films are not relaxor ferroelectrics but consist of PNRs, a more detailed description of the coexistence of both ferroelectric and relaxor features is necessary.

2) Relaxor behavior, e.g. the presence of PNRs, was initially discovered already in 1954 even in $\text{BaTiO}_3\text{-BaSnO}_3$ solid solutions [G.A. Smolenskii, V.A. Isupov, Segnetoelektricheskie svoistva tverdykh rastvorov stannata bariya v titanate bariya, Zh. Tekh. Fiz. 24 (1954) 1375–1386 (in Russian)]. This was not acknowledged by the authors. In such solid solutions $\text{BaTi}_{1-x}\text{Sn}_x\text{O}_3$, relaxor behaviour starts from about $x = 0.3$ and continues up to the solution limit [DOI: 10.1063/1.1829794 10.1080/01411590802457888]. The lower limit is attributed to the establishment of a continuous chain of Sn-O-Sn bonds. Below this limit, coexistence of both ferroelectric and relaxor behaviour was obtained [DOI: 10.1080/01411590802457888]. The authors prepared thin films from a $\text{BaTi}_{0.85}\text{Sn}_{0.15}\text{O}_3$ precursor solution. However, they do not provide a composition of their films. Dielectric behaviour in this work is probably similar to the one of ceramic samples with $x = 0.1, 0.15$ and 0.175 in [DOI: 10.1080/01411590802457888] exhibiting coexistence of ferroelectric and relaxor behaviour. To confirm this, the authors should provide more detailed data on dielectric dispersion around the temperature of maximum dielectric permittivity.

3) The authors have used several experimental techniques (HRTEM; PFM, Raman scattering) to obtain PNRs. The typical size of polar nanoclusters is about 2-10 nm [DOI:10.1039/D4TA00240G, 10.1142/S2010135X1241010X] in good agreement with HRTEM data of this work (row 220-222). Thus PFM is not an adequate mean for PNR analysis since it allows to visualize only mesoscopic domain patterns. On the other hand, relaxor ferroelectrics exhibit characteristic labyrinthine-like

domain structures [DOI:10.1146/annurev-matsci-071312-121632] which were not observed in this work. The presence of PNRs leads to symmetry breaking and thus broadening the Raman peaks compared to the ones in conventional ferroelectrics. In this work, Raman peak width was not analysed. Here, peak broadening was solely attributed to sintering temperature (row 198-199). Also, modes that are normally inactive or silent in a perfect crystal can become active due to local distortions introduced by PNRs. No such anomalous modes were obtained (may be the modes at 300 cm⁻¹ and 730 cm⁻¹?).

4) In relaxor ferroelectrics, a temperature T^* marks the crossover from the occurrence of both static and dynamic polar PNRs to solely dynamic ones [DOI: 10.1103/PhysRevB.72.184106, DOI: 10.1103/PhysRevB.80.064103]. Consequently, below T^* losses attributed to static PNRs become an issue for microwave applications. This issue should be considered by the authors.

Reviewer #2

(Remarks to the Author)

This manuscript reports on tin-doped barium titanate ($\text{BaTi}_{1-x}\text{Sn}_x\text{O}_3$, BTS) thin films prepared via a sol-gel method, and systematically investigates the effect of sintering temperature on their structural, microstructural, and dielectric properties. The argumentation is clear and the experimental design is reasonable. However, before this manuscript can be considered for acceptance in such a high-quality journal, the following points need to be further clarified.

1. Regarding the XRD patterns in Fig. 1a-c: Could the authors please explain the origin of the broad, amorphous-like hump observed at around 20° in all patterns?
2. Regarding Lines 89-91: The authors attribute the increase in lattice parameter with sintering temperature to grain growth and the relaxation of internal stress. This explanation is plausible, but for a more comprehensive discussion, should the potential influence of other factors, such as a change in oxygen vacancy concentration, also be considered and briefly discussed?
3. Regarding Fig. 2b: The dielectric loss of the BTS900 sample exhibits an inflection point (or a shoulder) around 100 kHz, which is distinct from the smoother curves of BTS850 and BTS800. Could the authors provide a more detailed explanation for this anomalous loss behavior, possibly from the perspective of its unique microstructure (e.g., larger grains) and a potentially higher concentration of oxygen vacancies?
4. The manuscript repeatedly emphasizes the trade-off between high tunability and low dielectric loss in both the abstract and introduction, claiming that the present work achieves "minimal losses." However, in the results section, particularly in Fig. 2f which showcases the microwave performance, only the tunability data is provided, while the corresponding microwave dielectric loss ($\tan\delta$) is completely missing. It is recommended that the authors add a companion figure next to Fig. 2f, presenting the dielectric loss ($\tan\delta$) of the BTS850 film in the 1-6 GHz frequency range under various DC biases. Furthermore, the Figure of Merit (FoM) should be calculated and discussed, and a comparison with other high-performance materials from the literature should be made to truly substantiate the claimed "superiority" of the material.
5. Regarding the TEM images in Fig. 4f-g: The images of the nanoclusters appear somewhat indistinct, making it difficult to clearly observe the clusters. Could the authors provide additional magnified images of other representative nanoclusters from Fig. 4e in the Supplementary Materials to strengthen their claim?
6. The experimental findings could be further strengthened by theoretical calculations (e.g., using DFT) to model the effect of Sn substitution on the lattice stability and the formation energy of polar nanoclusters. While this is beyond the scope of the current manuscript, the authors are encouraged to consider it for future studies.
7. Regarding Fig. 2c: It is recommended to change the x-axis from Voltage (V) to Electric Field (kV/cm) to facilitate standardized comparisons with other works.
8. There are several typographical errors in the text (e.g., in Line 140 and Line 198) that need to be corrected.

Reviewer #3

(Remarks to the Author)

The authors demonstrate the growth of BTS thin films using the sol-gel method, tune their dielectric properties, and explore the mechanism. However, several questions need to be addressed before this article can be considered for publication.

1. Page 4: What are the roughness values and grain size for BTS800, BTS850, and BTS900 films? Quantifying these values will help in the discussion
2. Page 4: EDX mapping of the cross-sectional samples along with the SEM images will help in explaining the interdiffusion in the samples. It might be difficult to conclude from the SEM images alone.
3. All the captions in the Supplementary Information are not formatted properly.
4. According to the authors, BTS900 shows significant interdiffusion between the film and Pt electrode which will degrade its dielectric properties. Still, it is showing the highest permittivity. Why? EDX images of the cross-sectional samples, showing the extent of interdiffusion, might help in the understanding.
5. At what voltage and at what frequency are the measurements of Fig. 2b and Fig. 2c carried out?
6. What are thickness of the thin films. In case of interdiffusion, how is the thickness calculated to estimate the electric field in Fig S2a-c

7. Figure 4: How do the actual SAED pattern of the sample look like? FFT of 'b' and 'c' regions look interchanged. Further, 'c' part of the sample doesn't look focused and that might lead to additional spots. In addition to the white arrows that show additional spots, there is a whole row of spots, which would signify another row of periodically arranged atoms.

Reviewer #4

(Remarks to the Author)

This article is about the microwave tunable thin films where well-known Sn doped BTO is study. I do not think that this work is appropriate to be considered for publication in this well-reputed journal. Apart from that I have a few of the following questions:

- Material chosen can be additionally characterized and may be synthesis by other techniques to overcome extrinsic factors.
 - Pure BTO is tetragonal (P4mm structure), however, with doping lattice constants are same which should decrease the polarization significantly, therefore its better to prepare sample with polar tetragonal grains to acquire better results. Therefore, materials choice and morphological design are required to be revisited.
 - Pores are evident from the surface and cross-sectional SEM images which are bound to give rise to extrinsic contributions in the electrical response such as polarization and dielectric constant.
 - The quality of SEM images (figure S1) is poor. There seems to be amorphous regions and grain boundaries which should give rise to space charge polarization. These contributions must be identified and explained in the PE loop and dielectric sections. It may also require detailed impedance spectroscopy for this purpose.
 - A weak hump in the dielectric constant has to be appropriately explained. What was the temperature at which XRD data was recorded? Then what was the transition in the dielectric data arises in correlation with the structural analysis.
 - There seems to be a role of conductivity in the PE loops and IE curves. This is evident from the broadness of IE switching peak and curvature at the high field PE data. This is required to be addressed using PUND data or from the fitting of IV data.
 - As per authors claim if there are polar nanoregions then the system should exhibit relaxor like features. Where is the discussion about this? Frequency and temperature dependent dielectric profile is to be reported for this.
 - In the explanation of figure 3h, authors claimed about long range order, that has to be proved from the PFM data.
 - Curve fitting in figure S5 requires a greater number of particles due to the statistical reason. Particle count is low for 50 to 60nm. Similarly, domain size distribution is not uniform. This raises the question about the region of the samples chosen for the measurement.
 - Raman peak fitting seems to accommodate more peaks. The broadness of the peak in terms of defects is to be explained.
 - Circuit analysis in figure S8 does not include polydispersivity which is unavoidable unless sample is single crystal or epitaxially grown.
 - Instead of using microprobe, contact with wire bonding are better to avoid any contact related contribution in the electrical response.
 - If Pt is used as bottom electrode, then electrical measurement can be performed in the perpendicular configuration instead of surface specific electrodes.
 - What happens to Pt diffusion into the film interface at such a high temperature? It's better to prepare Perovskite thin films on substrate such as STO, if high temperature annealing is required. The substrates should have the same crystal structure of the film to be deposited and lesser diffusion related issues due to their high temperature stability.
 - Choice of dopants which decreases critical temperatures and brings them near the room temperature, is not appropriate. For the applications materials with higher Tc are required for the thermal stability of the polar regions. In order to address these samples with pure BTO without any doping should be prepared for comparison as they are better ferroelectric.
 - How are the inset of figure 4 h constructed? On which basis are extreme polar configurations shown as the temperature range is narrow?
 - There must be more detail and further experiments on the choice of electrode geometry. This may require separate study to be reported. As claimed by the author, how does this geometry minimize the parasitic effect. These points are not clear in the text.
 - Resonant peaks are associated with the Piezoelectric effect. Why so, how to prove it? For this Piezoelectric response of the films is to be presented with the same electrode geometry as used in the manuscript.
- On the basis of above points, I am rejecting the article.

Version 1:

Reviewer comments:

Reviewer #1

(Remarks to the Author)

The authors have mostly taken my comments into account. However, for readers understanding and fitting with the scientific community, I recommend the authors to emphasize in the introduction the coexistence of ferroelectric and relaxor features in the crossover region $0.175 < x < 0.25$ following [<https://doi.org/10.1080/01411590802457888>] (note that the same case was described shortly in Ref. 16), to give a reference to this paper and to relate their samples to this crossover region where they can make use of the advantages of PNRs without losing too much tunability. Since the tunability of BST decreases with increasing Sn fraction [cf. Ref.17], the authors have to discuss the tradeoff between Sn content and tunability. Some misprints (in Ref. 17) and formatting mistakes in the reference list should be corrected.

Reviewer #2

(Remarks to the Author)

I thank the authors for their thorough and thoughtful revision of the manuscript.

They have successfully addressed all of my previous comments. The manuscript is much improved and is now a strong piece of work.

I have no further comments and am happy to recommend its publication in Nature Communications. Congratulations on the excellent work.

Reviewer #3

(Remarks to the Author)

Please accept the manuscript as is.

Responses to Editor and Referees' Comments

We thank the editor and referees for their helpful comments and respond to their specific points below. Changes to the manuscript are highlighted in the text.

Reviewer #1:

The experimental part of this paper is well done and well documented. However, the scientific part contains a conceptual shortage: 1) The aim of this work is described by the authors as following: "This work advances the understanding of polarization dynamics in BTS thin films and offers new strategies for optimizing tunable dielectric materials for microwave applications", "The remarkable performance (tunability) is attributed to the engineering of polar nanoclusters within the thin film matrix" and "This work demonstrates a promising strategy to achieve high tunability with minimal losses in ferroelectric thin films". On the other hand it is well known that "the tunability in relaxor ferroelectric exhibits a higher value as compared to normal ferroelectrics. It happens due to the presence of short-range ordering PNRs, which respond very quickly with the applied electric field compared to the long-range ordering of dipoles in normal ferroelectrics. [DOI: 10.5772/intechopen.96185]" Thus, the concept of this work is not new. Also, this work is not self-consistent concerning the presence of polar nanoregions (PNRs) and the based on these PNRs relaxor behavior:

1) Lets start with the definition of relaxors:"Relaxors are defined as ferroelectrics in which the maximum in the temperature dependence of static susceptibility occurs within the temperature range of dielectric relaxation, but does not coincide with the temperature of singularity of relaxation time or soft mode frequency" [DOI: 10.1142/S2010135X1241010X]. This means that a diffuse peak of dielectric permittivity and a typical frequency dispersion in the temperature range of the peak are characteristic features of ferroelectric relaxors. In this work, figures S3 do not show the sharp peak in the temperature dependence of dielectric permittivity known for relaxors. Also no dispersion of losses above temperature of maximum dielectric permittivity was obtained. Thus, nor dielectric permittivity nor losses show typical behaviour of relaxor ferroelectrics. Finally, the authors themselves exclude relaxor-type of ferroelectric behaviour (row 143-144). This also coincides with the fact that in figures S2a-c hysteresis curves are not the slim ones known for relaxor ferroelectrics. Once the fabricated films are not relaxor ferroelectrics but consist of PNRs, a more detailed description of the coexistence of both ferroelectric and relaxor features is necessary.

RE: We thank the reviewer for the valuable comments regarding the role of relaxor ferroelectrics in tunable dielectric applications. However, the strategy of this work is to develop non-relaxor type ferroelectrics for tunable applications in order to avoid the high dielectric loss associated with the dynamic polar nanoregions. In classical relaxor ferroelectrics, PNRs exhibit field dependent volume changes and broad frequency dependent dielectric dispersion, which increase loss compare to non-relaxor systems such as Sr-doped BaTiO₃, where its T_c is frequency independent and losses remain relatively low [*J. Am. Ceram. Soc.* 87(6), 1082–1087 (2004); *Nature Commun.*12, 3509 (2021); *J. Electroceram.* 11, 5–66 (2003)].

Additionally, the reference suggested by the reviewer [*Appl. Phys. Lett.* 104, 182910 (2014), *Appl. Phys. Lett.* 85, 5319–5321 (2004)] shows that in $\text{BaTi}_{1-x}\text{Sn}_x\text{O}_3$ solid solutions, relaxor behaviour appears only for $x \geq 0.175$, whereas for $x < 0.175$, the dielectric peak temperature T_c remains frequency independent. Our BTS films ($x = 0.15$) lie this non-relaxor regime, consistent with our experimental observation in Fig. S4, where T_c remains unchanged across frequencies, confirming the absence of classical relaxor behaviour.

Importantly, the high tunability in our BTS films originates from dynamic polar nanoclusters in the paraelectric phase with fixed T_c . These clusters have very short relaxation times, allowing rapid response without contributing significantly to dielectric loss. [*Phys. Chem. B*, 114, 49, 16465–16470 (2010); *Nature Commun.* 12, 3509 (2021)] This mechanism provides strong tunability while avoiding the drawbacks of relaxor behaviour. Moreover, it has been reported that ‘The change from normal ferroelectric into relaxor ferroelectric had a negative impact on the tunability value of the materials’ [*Appl. Phys. Lett.* 85, 5319–5321 (2004)]. We now have modified content in the revised manuscript to emphasize these points.

Revised version:

‘Compare with $\text{Ba}_{1-x}\text{Sr}_x\text{TiO}_3$, $\text{BaTi}_{1-x}\text{Sn}_x\text{O}_3$ exhibits superior dielectric tunability at radio frequency, primarily due to the greater chemical stability of Sn compared to Ti ion. Previous studies have shown the Sn doped barium titanate exhibits normal ferroelectric for $x \leq 0.175$ and transform to relaxor ferroelectric state for $x \geq 0.30$, respectively. The transition from a normal ferroelectric to relaxor ferroelectric phase leads to a decrease in tunability value in the materials. In particular, $\text{BaTi}_{0.85}\text{Sn}_{0.15}\text{O}_3$ exhibits a Curie point near room temperature, making it highly suitable for practical microwave tunable applications.’

‘Moreover, the frequency independence of T_c of BTS thin films indicates the absence of relaxor type of ferroelectric behaviour. It’s worth noting that barium titanate based relaxor ferroelectrics (such as $\text{BaTi}_{0.90}\text{Ga}_{0.05}\text{O}_3$ and $\text{BaTi}_{0.70}\text{Zr}_{0.30}\text{O}_3$) also exhibit high tunability, but they suffer higher microwave loss than the normal ferroelectrics such as $\text{Ba}_{0.6}\text{Sr}_{0.4}\text{TiO}_3$. Compared to the BTS800, the BTS850 thin film exhibits a T_c closer to room temperature, resulting in a higher concentration of polar nanoclusters and enhanced dielectric tunability.’

2) Relaxor behavior, e.g. the presence of PNRs, was initially discovered already in 1954 even in BaTiO_3 - BaSnO_3 solid solutions [G.A. Smolenskii, V.A. Isupov, *Segnetoelektricheskie svoistva tverdykh rastvorov stannata bariya v titanate bariya*, *Zh. Tekh. Fiz.* 24 (1954) 1375–1386 (in Russian)]. This was not acknowledged by the authors. In such solid solutions $\text{BaTi}_{1-x}\text{Sn}_x\text{O}_3$, relaxor behaviour starts from about $x = 0.3$ and continues up to the solution limit [DOI: 10.1063/1.1829794 10.1080/01411590802457888]. The lower limit is attributed to the establishment of a continuous chain of Sn-O-Sn bonds. Below this limit, coexistence of both ferroelectric and relaxor behaviour was obtained [DOI: 10.1080/01411590802457888]. The authors prepared thin films from a $\text{BaTi}_{0.85}\text{Sn}_{0.15}\text{O}_3$ precursor solution. However, they do not provide a composition of their films. Dielectric behaviour in this work is probably similar to the one of ceramic samples with $x = 0.1, 0.15$ and 0.175 in [DOI: 10.1080/01411590802457888] exhibiting coexistence of ferroelectric and relaxor behaviour. To confirm this, the authors should provide more detailed data on dielectric dispersion around the temperature of maximum dielectric permittivity.

RE: Following the reviewer's recommendation, we have now cited the suggested reference [G.A. Smolenskii, V.A. Isupov, Segnetoelektricheskie svoistva tverdykh rastvorov stannata bariya v titanate bariya, Zh. Tekh. Fiz. 24 (1954) 1375–1386 (in Russian)] in the introduction on page 3. We also cited the APL paper [DOI: 10.1063/1.1829794], which reports that 'The change from normal ferroelectric into relaxor ferroelectric had a negative impact on the tunability value of the materials.', reinforcing our selection of $x = 0.15$ to achieve high tunability. As shown in Fig. S4 in revised SI file, BTS shows a frequency independent Curie temperature, in agreement with with the previous studies and confirming its non relaxor behaviour.

3) The authors have used several experimental techniques (HRTEM; PFM, Raman scattering) to obtain PNRs. The typical size of polar nanoclusters is about 2-10 nm [DOI:10.1039/D4TA00240G, 10.1142/S2010135X1241010X] in good agreement with HRTEM data of this work (row 220-222). Thus PFM is not an adequate mean for PNR analysis since it allows to visualize only mesoscopic domain patterns. On the other hand, relaxor ferroelectrics exhibit characteristic labyrinthine-like domain structures [DOI:10.1146/annurev-matsci-071312-121632] which were not observed in this work. The presence of PNRs leads to symmetry breaking and thus broadening the Raman peaks compared to the ones in conventional ferroelectrics. In this work, Raman peak width was not analysed. Here, peak broadening was solely attributed to sintering temperature (row 198-199). Also, modes that are normally inactive or silent in a perfect crystal can become active due to local distortions introduced by PNRs. No such anomal modes were obtained (may be the modes at 300 cm^{-1} and 730 cm^{-1} ?).

RE: We agree with the reviewer that the resolution of PFM is not sufficient to directly observe polar nanoclusters in this work. That is why we used HRTEM to confirm the presence of the polar nano clusters. PFM in our study was used to show the different structural and phase variations with temperatures at above and below T_c . As we explained above, the materials in this work are ferroelectrics rather than relaxor ferroelectrics, such labyrinthine structures are not expected to be observed. In the Raman data, we did observe weak peaks at ≈ 300 and 730 cm^{-1} (marked in the Fig. 3f), modes that are forbidden in an ideal cubic phase, which we attributed to locally tetragonal polar nanoclusters embedded in the cubic matrix. As shown in Fig. 3g, the BTS800 film exhibits slightly broader peaks at ≈ 520 and 730 cm^{-1} compared to BTS 850 and BTS900, indicating less coherent local polar ordering at lower sintering temperatures.

4) In relaxor ferroelectrics, a temperature T^ marks the crossover from the occurrence of both static and dynamic polar PNRs to solely dynamic ones [DOI: 10.1103/PhysRevB.72.184106, DOI: 10.1103/PhysRevB.80.064103]. Consequently, below T^* losses attributed to static PNRs become an issue for microwave applications. This issue should be considered by the authors.*

RE: As we explained above and supported by the reviewer recommended references. The frequency independent of the Curie temperature confirm their normal ferroelectric nature in order to achieve high tunability and low loss. As such, the presence of a T^* temperature and associated microwave losses from static PNRs are not relevant to our system.

Reviewer #2:

This manuscript reports on tin-doped barium titanate ($BaTi_{1-x}Sn_xO_3$, BTS) thin films prepared via a sol-gel method, and systematically investigates the effect of sintering temperature on their structural, microstructural, and dielectric properties. The argumentation is clear and the experimental design is reasonable. However, before this manuscript can be considered for acceptance in such a high-quality journal, the following points need to be further clarified.

1、 Regarding the XRD patterns in Fig. 1a-c: Could the authors please explain the origin of the broad, amorphous-like hump observed at around 20° in all patterns?

RE: The broad, amorphous-like bump observed at around 20° originates from the substrate, which exhibits diffuse scattering at this angle. As shown in the fitted profile of the pure substrate below, the same feature is present. The sharp diffraction peaks in the patterns correspond to Pt electrode and Si substrate.

2、 Regarding Lines 89-91: The authors attribute the increase in lattice parameter with sintering temperature to grain growth and the relaxation of internal stress. This explanation is plausible, but for a more comprehensive discussion, should the potential influence of other factors, such as a change in oxygen vacancy concentration, also be considered and briefly discussed?

RE: We thank the reviewer for this insightful suggesting. We agree that oxygen vacancy concentration can influence the lattice parameter. To investigate this possibility, we have analysis the XPS spectra of Ba, Sn, Ti and O for the studied films. As shown in the revised Fig.1g. Ba and Sn spectra exhibit noticeable shoulders in BTS800, which significantly reduced in BTS850 and BTS900. These shoulders likely originate from residual organic/inorganic precursors that were not fully decomposed at lower sintering temperature. This is consistent with the SEM observations, where only BTS800 shows amorphous morphology surrounding the defined grains. The O 1s spectra of BTS800 exhibit a more intense higher binding energy

component at 531-532eV, which is typically attributed to the defect related oxygen species such as oxygen vacancies, absorbed oxygen or remaining of incomplete precursor decomposition. In contrast, this component is notably suppressed in BTS850 and BTS900, suggesting a reduction in oxygen related surface or defect species with increasing sintering temperature rather than oxygen vacancies. Moreover, we note that the primary peaks of all the chemical elements remains at similar binding energies for all samples, without significant chemical shifts. Such stable main peak positions indicates that the bulk chemical states remain unchanged and suggest oxygen vacancies induced by the sintering temperature are minor. While we cannot completely rule out a minor contribution from oxygen vacancies, their influence on the lattice parameter is limited. Therefore, the observed increase in lattice parameter with sintering temperature is primarily attributed to grain growth and relaxation of internal stress rather than changes in oxygen vacancies. We have now modified the manuscript accordingly.

Revised version:

'The X-ray photoelectron spectroscopy (XPS) analysis reveals pronounced shoulder features in the Ba 3d and Sn 3d spectra of BTS800, indicating the residual component from incomplete precursor decomposition. These shoulders are notably decreases in BTS850 and BTS900, consistent with SEM observations of amorphous regions in BTS800 and the well defined grains at higher sintering temperatures. The O 1s spectra show a more intense defect related peak at ca. 531.5 eV in BTS800, attributed to oxygen vacancies or absorbed oxygen components, which decreases with increasing sintering temperature. Notably, no clear binding energy shifts are observed in the main peaks of Ti, Sn, O and Ba indicating stable oxidation states and minimal influence of oxygen vacancies.'

3、 *Regarding Fig. 2b: The dielectric loss of the BTS900 sample exhibits an inflection point (or a shoulder) around 100 kHz, which is distinct from the smoother curves of BTS850 and BTS800. Could the authors provide a more detailed explanation for this anomalous loss behavior, possibly from the perspective of its unique microstructure (e.g., larger grains) and a potentially higher concentration of oxygen vacancies?*

RE: We thank the reviewer for the careful observation. The inflection point in the dielectric loss curve in BTS900 around 100kHz is likely attribute to dielectric relaxation associated with dipole polarization. This behaviour is consistent with its larger grain size, reduced grain boundary density and a Curie point above room temperature, allowing dipoles to remain active and relax in this frequency range. In contrast, BTS800 and BTS850, with lower Curie points, do not exhibit such features. XPS results show no strong evidence of increase oxygen vacancies in BTS900 to supporting the dielectric loss anomaly. We now have revised the manuscript to clarify this point.

Revised version:

'Dielectric loss decreases with increased frequency in Fig. 2b, primarily due to the suppression of the extrinsic loss mechanisms at high frequencies. Around 100kHz, the dielectric loss exhibits a notable drop with increasing frequency, which can be attributed to the dielectric relaxation associated with dipole polarization.²⁶ Since the T_c temperature of BTS900 was slightly above room temperature (Fig. S4), dipolar contribution remain active and relax

around this frequency range. In contrast, BTS800 and BTS850, with their T_c below room temperature, exhibit no such anomaly, as their dipoles are less active. The drop in loss at high frequencies likely results from the diminished contribution of dipolar polarization.'

4、 *The manuscript repeatedly emphasizes the trade-off between high tunability and low dielectric loss in both the abstract and introduction, claiming that the present work achieves "minimal losses." However, in the results section, particularly in Fig. 2f which showcases the microwave performance, only the tunability data is provided, while the corresponding microwave dielectric loss ($\tan\delta$) is completely missing. It is recommended that the authors add a companion figure next to Fig. 2f, presenting the dielectric loss ($\tan\delta$) of the BTS850 film in the 1-6 GHz frequency range under various DC biases. Furthermore, the Figure of Merit (FoM) should be calculated and discussed, and a comparison with other high-performance materials from the literature should be made to truly substantiate the claimed "superiority" of the material.*

RE: We apologize for the missing of the dielectric loss data. We have now included the microwave dielectric loss data in Fig. 2e, which shows that the loss value in the range of 0.027 to 0.020 across the 1-6GHz frequency range. However, the dielectric loss under DC bias field remain changeling for thin film sample and we could not accurately extract their loss data under electric field. To address the reviewer's suggestion regarding the Figure of Merit (FoM), we have now added a comparative Table S4 that compare our results against other high performance tunable dielectric thin films. The BTS850 film shows a FoM of 37 at 6GHz, which exceeds that most comparable systems, highlighting our claim of high tunability with low loss. The manuscript has been revised accordingly in include and discuss these results.

Revised version:

'The dielectric loss of BTS850 was found to be *ca.* 0.027 to 0.020 across the 1-6GHz frequency range, which is comparable to the other tunable thin films^{26,38-42} (Table S4). The dielectric permittivity decreased uniformly across the frequency range under electric field. Additional, resonant peaks were only observed at 2.3GHz and 4.8GHz under bias fields, attributed to piezoelectric effect⁴³⁻⁴⁶, where aligned dipoles induce resonance phenomena at specific frequencies. Excluding these resonance regions, the BTS850 thin film exhibit a high tunability of *ca.* 74 % under a relatively low applied DC bias of 15 V (equivalent of 273kV cm⁻¹) with a low loss of 0.020, yielding a figure of merit (FoM) of 37 at 6GHz (Table S4).'

5、 *Regarding the TEM images in Fig. 4f-g: The images of the nanoclusters appear somewhat indistinct, making it difficult to clearly observe the clusters. Could the authors provide additional magnified images of other representative nanoclusters from Fig. 4e in the Supplementary Materials to strengthen their claim?*

RE: We have added new Figures (Fig. 4d-g and Fig. S8) showing enlarged images of regions in BTS850 grain having various distortions and defects ("nanopolar" regions, structural modulations and dislocations).

6、 *The experimental findings could be further strengthened by theoretical calculations (e.g., using DFT) to model the effect of Sn substitution on the lattice stability and the formation*

energy of polar nanoclusters. While this is beyond the scope of the current manuscript, the authors are encouraged to consider it for future studies.

RE: We sincerely appreciate the reviewer's insightful suggestion regarding the use of DFT calculations to model the effect of Sn substitution on lattice stability and polar nanocluster formation. We fully agree that such theoretical analysis would provide valuable complementary insights. However, as our current expertise and resources are focused on experimental investigation, incorporating reliable DFT calculations is beyond the scope of this work. Nevertheless, we acknowledge the importance of this approach and will actively consider collaborating with theoretical groups or developing the necessary computational capabilities for future studies.

7、 *Regarding Fig. 2c: It is recommended to change the x-axis from Voltage (V) to Electric Field (kV/cm) to facilitate standardized comparisons with other works.*

RE: We thank the reviewer for the suggestion to re-plot Fig. 2c against electric field. In the current work, one of main objective is to highlight the low voltage required to achieve high tunability in BTS film, which is an attribute of direct relevance to on-chip integration. All three sample share similar thickness ca. 500nm, which can readily convert voltage to field (A DC bias of 1 V corresponding to an electric field of ca. 2 kV cm^{-1}) if desired. To facilitate this, we have now added a sentence to the Fig. 2c caption giving the field equivalence, but we have retained the voltage axis to keep the focus on low bias operation that distinguishes our films from bulk materials.

Revised version :

Fig. 2 Dielectric properties of the BTS thin films. Schematic of the dielectric tunability measurement setup covering both (a) low and (d) high frequency ranges. b frequency dependence of dielectric permittivity and loss tangent for studied BTS films in the low frequency range under a 0.5 V AC voltage. Electric field dependent of c dielectric tunability for BTS films measured at 1 MHz. e Dielectric permittivity and f tunability for the BTS850 film in the high frequency range. convert voltage to field. (note: A DC bias of 1 V corresponding to an electric field of ca. $2\text{ kV}\cdot\text{cm}^{-1}$)'

8、 *There are several typographical errors in the text (e.g., in Line 140 and Line 198) that need to be corrected.*

RE: We thank the reviewer for the careful review. The typographical errors have now been corrected.

Reviewer #3:

The authors demonstrate the growth of BTS thin films using the sol-gel method, tune their dielectric properties, and explore the mechanism. However, several questions need to be addressed before this article can be considered for publication.

1. Page 4: What are the roughness values and grain size for BTS800, BTS850, and BTS900 films? Quantifying these values will help in the discussion

RE: We thank the reviewer for the helpful comment. In response, we have now included the quantified grain size and surface roughness values for the BTS800, BTS850, and BTS900 thin films in the revised manuscript. As shown in Supplementary Fig. 1, the average grain sizes are 23.6 ± 5.2 nm for BTS800, 54.1 ± 14.5 nm for BTS850, and 69.5 ± 16.9 nm for BTS900, indicating grain growth with increasing sintering temperature. The corresponding surface roughness (Rq) values, obtained from AFM measurements, are 3.77 nm, 5.02 nm, and 5.08 nm, respectively, showing an increase in surface roughness. These results have been added to the revised text.

Revised version:

‘which promotes structural densification and integrity. A few isolated pores are observed, but given their low concentration and large permittivity contrast between air ($\epsilon' \approx 1$) and BTS ($\epsilon' \approx 600$), the effect on the overall dielectric constant is expected to be very minor. The surface morphology images show the average grain size increase from 23.6 ± 5.2 nm (BTS800) to 69.5 ± 16.9 nm (BTS900), with a rise in root-mean-square surface roughness (Rq) from 3.77 nm to 5.08 nm (Fig. S1). The BTS800 film exhibits a smallest grain structure,’

2. Page 4: EDX mapping of the cross-sectional samples along with the SEM images will help in explaining the interdiffusion in the samples. It might be difficult to conclude from the SEM images alone.

RE: We thank the reviewer for this valuable suggestion. We have now added cross-sectional EDX maps in Fig. S2 alongside the corresponding SEM images, to show the interface between the film and electrode in BTS800, BTS850 and BTS900 samples. Additionally, HAADF image combined with EDX mapping (Fig. S9) for BTS850 reveals an interface roughness of *ca.* 15 nm, but no significant diffusion between film and electrode was observed.

3. All the captions in the Supplementary Information are not formatted properly.

RE: We thank the reviewer for pointing out the formatting issues. All captions in the Supplementary Information have now been reformatted according to the journal’s guidelines.

4. According to the authors, BTS900 shows significant interdiffusion between the film and Pt electrode which will degrade its dielectric properties. Still, it is showing the highest permittivity. Why? EDX images of the cross-sectional samples, showing the extent of interdiffusion, might help in the understanding.

RE: We thank the reviewer for this insightful comment. BTS900’s highest permittivity arises primarily from significantly increased grain size and enhanced local polar ordering at the higher sintering temperature. Larger grains reduce grain-boundary density and internal stress, leading to higher intrinsic polarizability and thus higher overall permittivity. Additionally, the Curie temperature (T_c) of BTS900 is slightly above room temperature (Fig. S4), allowing extra dipolar contributions to remain active at measurement temperatures, further enhancing the dielectric permittivity but also increasing dielectric loss.

Revised version:

'Among the samples, BST900 shows the highest permittivity, followed by BTS850 and BTS800, consistent with enhanced grain growth and improved crystallinity observed in both SEM, EDX and XRD analyses. The larger grains in BTS900 reduce grain-boundary density and relieve internal stresses, enabling more rapid dipole response and consequently, higher dielectric permittivity..... Since the T_c temperature of BTS900 was slightly above room temperature (Fig. S4), dipolar contribution remain active and relax around this frequency range. In contrast, BTS800 and BTS850, with their T_c below room temperature, exhibit no such anomaly, as their dipoles are less active.'

5. At what voltage and at what frequency are the measurements of Fig. 2b and Fig. 2c carried out?

RE: The applied ac voltage is 0.5V in Fig. 2b and the frequency in Figure 2c is 1MHz. We have now made them clear in the revised figure caption.

Revised version:

'**Fig. 2 Dielectric properties of the BTS thin films.** Schematic of the dielectric tunability measurement setup covering both (a) low and (d) high frequency ranges. b frequency dependence of dielectric permittivity and loss tangent for studied BTS films in the low frequency range under a 0.5 V AC voltage. Electric field dependent of c dielectric tunability for BTS films measured at 1 MHz. e Dielectric permittivity and f tunability for the BTS850 film in the high frequency range. convert voltage to field. (note: A DC bias of 1 V corresponding to an electric field of ca. $2 \text{ kV}\cdot\text{cm}^{-1}$)'

6. What are thickness of the thin films. In case of interdiffusion, how is the thickness calculated to estimate the electric field in Fig S2a-c.

RE: The thickness of our thin films, measured from cross-sectional SEM images in Fig. 1, is approximately 500–550 nm, yielding an applied electric field of about $2 \text{ kV}\cdot\text{cm}^{-1}$ per volt (note: A DC bias of 1 V corresponding to an electric field of ca. $2 \text{ kV}\cdot\text{cm}^{-1}$). Regarding interdiffusion, based on cross-sectional EDX maps (Fig. S2), the interdiffusion zone between Pt and the BTS films is typically limited to within a few tens of nanometres, which is negligible compared to the total film thickness. Additionally, HAADF image combined with EDX mapping (Fig. S9) for BTS850 reveals an interface roughness of ca. 15nm, but no significant diffusion between film and electrode was observed. We have now add the equivalent electric field in the Fig. 2 caption.

7. Figure 4: How do the actual SAED pattern of the sample look like? FFT of 'b' and 'c' regions look interchanged. Further, 'c' part of the sample doesn't look focused and that might lead to additional spots. In addition to the white arrows that show additional spots, there is a whole row of spots, which would signify another row of periodically arranged atoms.

RE: We thank the reviewer for the careful evaluation, and apologies indeed for the confusion caused by swapping images 'b' and 'c' in Figure 4. The 'c' part of the figure is in focus, however it shows an example of local deviation of BTS structure from pseudo-cubic one. A bright field

STEM images is presented in additional (Fig. 4d, 4e and Fig. S8) as well as corresponding selected area diffraction pattern (SAED) in Fig 4f and 4g. The discussion in the manuscript has been modified accordingly.

Revised version:

'This displacement is indicative of an incommensurate phase induced by local lattice distortions and is associated with the formation of polar nanoclusters. Local dislocations were observed in high resolution HAADF and ADF images (Fig. S8a and b) indicating nanoscale heterogeneity within the BTS850 matrix and suggesting the present of polar nanoclusters. Additionally, HAADF images across different regions and their corresponding selected area electron diffraction (SAED) patterns (Fig. 4d-g, Fig S8c and d) exhibit some additional diffraction spots, further supporting the existence of local structural distortions and short - range ordering in the system.'

Reviewer #4:

This article is about the microwave tunable thin films where well-known Sn doped BTO is study. I do not think that this work is appropriate to be considered for publication in this well-reputed journal. Apart from that I have a few of the following questions:

- *Material chosen can be additionally characterized and may be synthesis by other techniques to overcome extrinsic factors.*

RE: We appreciate the reviewer's suggestion to further strengthen our work through additional characterization and alternative synthesis routes. We have now carried out additional characterisation, including XPS and EDX to support our discussions. Although our current films were produced via sol-gel processing, we agree that techniques such as pulsed-laser deposition (PLD) or atomic-layer deposition (ALD) could further reduce extrinsic factors, such as porosity, grain-boundary density, and interdiffusion, and we plan to pursue these methods in future studies to optimize film quality and device performance.

- *Pure BTO is tetragonal (P4mm structure), however, with doping lattice constants are same which should decrease the polarization significantly, therefore its better to prepare sample with polar tetragonal grains to acquire better results. Therefore, materials choice and morphological design are required to be revisited.*

RE: The reviewer misunderstands the structure of dielectrics with high tunability and low loss. We would like to clarify that our material design intentionally departs from the tetragonal BaTiO₃ to achieve low loss and reversible tunability. Pure BaTiO₃ exhibits strong spontaneous and remanent polarization, which produces pronounced hysteresis and irreversible tuning, undesirable for microwave applications. By contrast, our BTS films adopt an average cubic structure with no long-range ferroelectric domains, yet retain local polar nanoclusters. This structure suppresses macroscopic hysteresis and remanent polarization, enabling reversible dielectric tuning under applied bias with minimal loss.

- Pores are evident from the surface and cross-sectional SEM images which are bound to give rise to extrinsic contributions in the electrical response such as polarization and dielectric constant.

RE: We agree that pores can introduce extrinsic contributions to the dielectric response. However, because the permittivity of air-filled pores (~ 1) is orders of magnitude lower than that of our BTS matrix (~ 600 at RF), even a simple effective-medium estimate [Lei Zhu, Phys. Chem. Lett. 5, 3677–3687 (2014)] shows that the impact of the low porosity observed in films is negligible. In practice, our dense microstructure, with only isolated pores visible by SEM, means that pore-related dielectric reduction is minimal. We have added a brief note in the revised manuscript to clarify this point.

Revised version:

'A few isolated pores are observed, but given their low concentration and large permittivity contrast between air ($\epsilon' \approx 1$) and BTS ($\epsilon' \approx 600$), the effect on the overall dielectric constant is expected to be very minor.'

- The quality of SEM images (figure S1) is poor. There seems to be amorphous regions and grain boundaries which should give rise to space charge polarization. These contributions must be identified and explained in the PE loop and dielectric sections. It may also require detailed impedance spectroscopy for this purpose.

RE: We now have changed SEM images in order to increase the quality of images. We thank the reviewer for raising the impact of space-charge polarization. As shown in the revised Fig.1g. Ba and Sn spectra exhibit noticeable shoulders in BTS800, which significantly reduced in BTS850 and BTS900. These shoulders likely originate from residual organic/inorganic precursors that were not fully decomposed at lower sintering temperature. This is consistent with the SEM observations, where only BTS800 shows amorphous morphology surrounding the defined grains. Space charge effects typically lead to pronounced low-frequency dispersion in ϵ' and indeed our BTS900 film shows a relatively strong drop in permittivity, which might attribute to space charge. In contrast, BTS800 and BTS850 exhibit weaker frequency dependence, indicating that space-charge contribution is negligible. We have modified manuscript to clarify the possible space-charge effects in BTS system.

Revised version:

'While the broadened P - E loops and tilted I - E loops suggest the presence of leakage current, possibly induced by the space charges, the displacement of switching current peaks from the maximum applied field indicates polar nanoclusters are responsible for the polarization behaviour in the film²⁷.'

- A weak hump in the dielectric constant has to be appropriately explained. What was the temperature at which XRD data was recorded? Then what was the transition in the dielectric data arises in correlation with the structural analysis.

RE: We believe the reviewer is asking the hump in Fig. S4a (revised version). The weak hump near -20°C is likely attributed to coexistence of multiple nanoscale polar phases below its Curie point. Similar low temperature hump behaviour has been reported in the nanograined

BaTiO₃ based ceramics and attributed to the multiple polar phases. [Zhao Z, Buscaglia V, Viviani M, et al (2004) Phys Rev B, 70, 241071; Lin S, Lü T, Jin C, Wang X (2006) Phys Rev B, 74,134116]. Our XRD scans were all collected at room temperature (*ca.* 20 °C), which is above the Curie point of the BTS800 film. We have now modified the manuscript to explain the weak dielectric hump corresponds the coexistence of mixed polar phases.

Revised version:

‘Unlike BTS bulk ceramics, which exhibit sharp dielectric permittivity peaks at the Curie point²⁶, the thin films show a broader permittivity peak, which gradual increase in dielectric permittivity upon heating(Fig. S4), followed by a subsequent decrease. A subtle anomaly near -20 °C is observed at *ca.* -20°C in the BTS800 sample, likely arising from the nanoscale coexistence of the multiple polar phases within nanograins^{29,30}. The observed T_c temperatures are 3.8 °C, 16.4 °C, 23.2 °C for BTS800, BTS850 and BTS900, respectively.’

- *There seems to be a role of conductivity in the PE loops and IE curves. This is evident from the broadness of IE switching peak and curvature at the high field PE data. This is required to be addressed using PUND data or from the fitting of IV data.*

RE: We appreciate the reviewer’s concern regarding conductivity contributions to our P-E and I-E measurement. Since our BTS films are tested around Curie temperature region, with dynamic polar nanoclusters rather than a typical ferroelectric switching behaviour. The standard PUND approach (designed to isolate 180° domain switching) is not directly applicable. We agree that conductivity does contribute to the observed broadening of switching peaks and the curvature at high fields. Nonetheless, the switching current peaks in I-E loops are not at the maximum applied field, indicating the dominant contributor to polarization behaviour in BTS film is polar nanoregions rather than conductivity. [Jin L, Li F and Zhang S, J. Am. Ceram. Soc., 97 [1] 1–27 (2014)]. We have now modified the manuscript accordingly.

Revised version:

‘While the broadened *P-E* loops and tilted *I-E* loops suggest the presence of leakage current, possibly induced by the space charges, the displacement of switching current peaks from the maximum applied field indicates polar nanoclusters are responsible for the polarization behaviour in the film²⁷. The sharpest current peaks are observed in BST900, further supporting its superior tunability.’

- *As per authors claim if there are polar nanoregions then the system should exhibit relaxor like features. Where is the discussion about this? Frequency and temperature dependent dielectric profile is to be reported for this.*

RE: We thank the reviewer for raising this point. As explained in our response to Reviewer 1, BTS films contain polar nanoclusters rather than polar nanoregions found in relaxor ferroelectrics. They show frequency independence of the permittivity peaks as shown in Fig. S4. By selecting $x = 0.15$, We intentionally preserve this non-relaxor behaviour to achieve high tunability without the increased dielectric loss associated with PNRs.

- In the explanation of figure 3h, authors claimed about long range order, that has to be proved from the PFM data.

RE: The long range ferroelectric order in the diagram is in fact demonstrated by our PFM data. Fig. 3a shows a uniform phase contrast image below T_c , with domain appearing multiple grains. Fig. 3b shows clear domain switching under ± 10 V electric field. Fig. 3c shows the phase and amplitude loops with sharp, square like switching, confirming the long range polarization. We have now modified the manuscript accordingly.

Revised revision:

'At 14 °C, which is below its Curie point T_c , the film exhibits a uniform phase contrast in the PFM image (Fig. 3a), with ferroelectric domains occurring multiple grains, indicative of well-defined and long-range ferroelectric ordering. Upon application of a DC bias of ± 10 V, clear domain switching is observed (Fig. 3b), further supporting existence of switchable polarization. The phase and amplitude hysteresis loops (Fig. 3c) show sharp, square-like switching behavior, characteristic of long-range ferroelectric domains.'

- Curve fitting in figure S5 requires a greater number of particles due to the statistical reason. Particle count is low for 50 to 60nm. Similarly, domain size distribution is not uniform. This raises the question about the region of the samples chosen for the measurement.

RE: We thank the reviewer for the suggestion. We have now increased both grain and domain measurements to over 100 counts to improve statistical confidence (Fig. S1 and Fig. S6). Regarding the non-uniform domain size distribution, ferroelectric thin films naturally exhibit a range of domain sizes due to local variations in strain, defect density and nucleation sites.

- Raman peak fitting seems to accommodate more peaks. The broadness of the peak in terms of defects is to be explained.

RE: The broadening of the Raman peaks in BTS films arises from several factors such as local lattice distortions, compositional fluctuations (Sn/Ti disorder) and oxygen vacancy related defect that accompany polar nanocluster formation. These structural heterogeneities increase the width of the Raman peaks. We have now modified the discussion to include the broadening of the Raman peaks [*J. Mater. Sci. Mater. Electron.* **34**, 1539 (2023); *J. Phys. D: Appl. Phys.* **42**, (2009)].

Revised version:

'Additionally, BTS800 shows slightly broader Raman peaks at the *ca.* 520 cm^{-1} and 730 cm^{-1} compare to BTS850 and BTS900 (Fig. 3g), which could arises from several factors such as local lattice distortion, local ferroelectric ordering involving Sn^{4+} and Ti^{4+} ions and residual impurity in the system^{49,50}.'

- Circuit analysis in figure S8 does not include polydispersivity which is unavoidable unless sample is single crystal or epitaxially grown.

RE: Polydisperse effects will appear as deviations of C_{center} and C_{outer} from ideal 90° capacitive phase. By labelling these elements as non-ideal capacitors (i.e. constant-phase elements), the schematic still captures any phase dispersion. Since Fig. S11 (in revised version) is intended simply to illustrate the topological correspondence between microstructure and circuit, rather than to serve as a fully parameterized lumped-element model for simulation, this notation is entirely appropriate. We now have added a note in the caption to clarify the point.

- *Instead of using microprobe, contact with wire bonding are better to avoid any contact related contribution in the electrical response.*

RE: We thank the reviewer for this suggestion. Our microprobe approach with multiple electrode sizes was chosen to minimize potential contact effects. By comparing dielectric response from different electrodes, we effectively isolate and remove contact contributions, focusing only on the intrinsic properties of the film. Furthermore, using a microprobe allows measurement near the film without introducing the additional parasitic inductance and capacitance associated with longer wire bonds.

- *If Pt is used as bottom electrode, then electrical measurement can be performed in the perpendicular configuration instead of surface specific electrodes.*

RE: We thank the reviewer for this suggestion. In fact, the electrical measurements in low frequency range (Fig. 2a-c) were carried out in the perpendicular (vertical) capacitor configuration. A Pt bottom electrode and a gold top electrode were used to apply the electric field across the film thickness.

- *What happens to Pt diffusion into the film interface at such a high temperature? It's better to prepare Perovskite thin films on substrate such as STO, if high temperature annealing is required. The substrates should have the same crystal structure of the film to be deposited and lesser diffusion related issues due to their high temperature stability.*

RE: We appreciate the reviewer's suggestion regarding the use of STO substrates to reduce interdiffusion during high-temperature sintering. We now have provided additional chemical maps collected from the cross-section TEM specimens (Fig. S9). The roughness of BTS850/Pt interface can be estimated of about 15 nm. The corresponding chemical mapping did not reveal significant diffusion of Pt into the BTS850 film. In addition, conductive bottom and top electrodes (Pt) are required to measure low-frequency tunability, dielectric permittivity and P-E/I-E loop measurements. As STO substrate is insulating, it cannot serve directly as a conductive electrode. Nevertheless, we agree that future studies exploring epitaxial growth of BTS thin films on STO substrates, possibly combined with conductive buffer layers such as LSMO or Nb-STO, would be valuable for further optimizing film quality and minimizing interdiffusion.

- *Choice of dopants which decreases critical temperatures and brings them near the room temperature, is not appropriate. For the applications materials with higher T_c are required for the thermal stability of the polar regions. In order to address these samples with pure BTO without any doping should be prepared for comparison as they are better ferroelectric.*

RE: We thank the reviewer for this comment. However, there appears to be a misunderstanding regarding the requirements for tunable dielectric applications. While pure BTO is ferroelectric at room temperature and suitable for piezoelectric or ferroelectric memory applications, its strong remanent polarization leads to significant hysteresis loss and requires high switching fields, making it unsuitable for low loss, reversible tunability. In contrast, materials in the paraelectric state, particularly near but above T_c , are preferred for tunable dielectrics due to their field sensitive permittivity, minimal hysteresis and low dielectric loss. Doping with Sn effectively lowers T_c to bring the system into paraelectric phase at room temperature, enabling efficient dielectric tuning under electric field. This strategy aligns with widely accepted approaches in tunable dielectric design [L.B. Kong, S. Li, T.S. Zhang, J.W. Zhai, F.Y.C. Boey, J. Ma, *Progress in Materials Science*, 55, 840–893(2010); A.K. Tagantsev, V.O. Sherman, K.F. Astafiev, J. Venkatesh, N. Setter, *Journal of Electroceramics*, 11, 5–66 (2003)]

• *How are the inset of figure 4 h constructed? On which basis are extreme polar configurations shown as the temperature range is narrow?*

RE: We thank reviewer for the observation and believe the comment refer to Fig. 3h, as there is no inset in Fig. 4h. The inset in Fig. 3h is intended to schematically illustrate the evolution of the polar structures with increasing temperature. while the experimental temperature range is relatively narrow, our dielectric data and PFM results indicate a well-defined T_c of 16.4 °C. Specifically, the diagram depicts the transition from long- range ordered ferroelectric domains below T_c to smaller, less stable polar cluster near T_c and ultimately to dynamic polar nanoclusters above T_c . This figure supports our interpretation of the temperature dependent dielectric behaviour. We have revised the manuscript to clarify this point more clearly.

Revised version:

‘Based on these findings, the evolution of polarization states with temperature in BTS850 is schematically summarized in Fig. 3h. The inset in Fig. 3h provides a qualitative visualization of this progression from ferroelectric domains to polar nanoclusters. Below T_c (point A), dipoles exhibit long-range ordered states,’

• *There must be more detail and further experiments on the choice of electrode geometry. This may require separate study to be reported. As claimed by the author, how does this geometry minimize the parasitic effect. These points are not clear in the text.*

RE: We thank the reviewer for this insightful comment. The parasitic effect in dielectric measurement is often linked to the geometry of the capacitor, particularly the diameter to thickness d/t ratio. As shown in the previous literature [Ding, S., Jia, J., Xu, B. *et al. Nat Commun* 16, 608 (2025).], parasitic edge effects become significant when the d/t ratio is <20 . In our study, the electrode diameter is ca. 400 μm and the film thickness is ca. 500nm, giving a d/t ratio of ca. 800. The continuous bottom electrode eliminates lateral leakage paths. This indicates that parasitic effect due to electrode geometry are negligible in our measurements. We have added a brief description of this design and its role in minimizing parasitic capacitance and leakage to the Methods section.

- *Resonant peaks are associated with the Piezoelectric effect. Why so, how to prove it? For this Piezoelectric response of the films is to be presented with the same electrode geometry as used in the manuscript.*

RE: We thank the reviewer for this comment. The observed peaks are not present at 0V and appear only under applied DC electric field, which suggests they are field induced electro-mechanical resonances (piezoelectric effect). These features are commonly observed in ferroelectric tunable dielectrics and arise from acoustic resonance modes. We now add more relevant references to support our discussion. [Gu, Z., Pandya, S., Samanta, A. *et al. Nature* 560, 622–627 (2018); V. Porokhonsky, J. Li, D. Damjanovic, *Appl. Phys. Lett.* 94, 212906 (2009); L. Wu, J. Wu, H. Huang, H. Bor, *Appl. Phys. Lett.* 90, 072901 (2007); X. Wei, Y. Feng, X. Yao, *Appl. Phys. Lett.* 84, 1534 (2004)]

On the basis of above points, I am rejecting the article.

RE: We thank the reviewer for their time and detailed feedback. We have carefully addressed all comments with appropriate physical explanations and supporting literature references. We believe the revisions and clarifications have substantially improved the manuscript and clarified the scope, methodology and significance of the work. We respectfully hope the reviewer will reconsider the manuscript in light of these comprehensive responses.

Responses to Editor and Referees' Comments

We thank the editor and referees for their helpful comments and respond to their specific points below. We have now revised our paper to address the editorial request according to the attached author checklist and the remaining comments from the reviewers. All Changes to the manuscript are highlighted in the text.

Reviewer #1:

The authors have mostly taken my comments into account. However, for readers understanding and fitting with the scientific community, I recommend the authors to emphasize in the introduction the coexistence of ferroelectric and relaxor features in the crossover region $0.175 < x < 0.25$ following [<https://doi.org/10.1080/01411590802457888>] (note that the same case was described shortly in Ref. 16), to give a reference to this paper and to relate their samples to this crossover region where they can make use of the advantages of PNRs without loosing too much tunability. Since the tunability of BST decreases with increasing Sn fraction [cf. Ref.17], the authors have to discuss the tradeoff between Sn content and tunability. Some misprints (in Ref. 17) and formatting mistakes in the reference list should be corrected.

RE: We thank the reviewer for the helpful comment. In response, we have now revised the introduction and clarified the rationale behind composition choice. Furthermore, we have corrected the formatting and minor errors in the reference list.

Revised version:

'Previous studies have shown the Sn doped barium titanate exhibits normal ferroelectric for $x \leq 0.175$ and transform to relaxor ferroelectric state for $x \geq 0.30$, respectively.^{16,17} In the intermediate crossover region ($0.175 < x < 0.25$), features of both ferroelectric and relaxor behaviours can coexist, attributed to the formation of polar nanoregions.¹⁸ As the Sn content increases, the transition from a normal ferroelectric to relaxor ferroelectric phase leads to a decrease in tunability value of materials.¹⁷ Notably, $\text{BaTi}_{0.85}\text{Sn}_{0.15}\text{O}_3$ exhibits a Curie point near room temperature^{19,20}, placing it just below the crossover region and making it highly suitable for practical microwave tunable applications.'

Reviewer #2:

I thank the authors for their thorough and thoughtful revision of the manuscript. They have successfully addressed all of my previous comments. The manuscript is much improved and is now a strong piece of work. I have no further comments and am happy to recommend its publication in Nature Communications. Congratulations on the excellent work.

RE: We thank reviewer for the kind words and thoughtful assessment of the manuscript.

Reviewer #3:

Please accept the manuscript as is.

RE: We thank reviewer for their time and effort in reviewing the manuscript.